# Adaptation decorrelates shape representations

Marcelo G. Mattar [1,5], Maria Olkkonen[2,3], Russell A. Epstein[1] & Geoffrey K. Aguirre [4]

Perception and neural responses are modulated by sensory history. Visual adaptation, an example of such an effect, has been hypothesized to improve stimulus discrimination by decorrelating responses across a set of neural units. While a central theoretical model, behavioral and neural evidence for this theory is limited and inconclusive. Here, we use a parametric 3D shape-space to test whether adaptation decorrelates shape representations in humans. In a behavioral experiment with 20 subjects, we find that adaptation to a shape class improves discrimination of subsequently presented stimuli with similar features. In a BOLD fMRI experiment with 10 subjects, we observe that adaptation to a shape class decorrelates the multivariate representations of subsequently presented stimuli with similar features in object-selective cortex. These results support the long-standing proposal that adaptation improves perceptual discrimination and decorrelates neural representations, offering insights into potential underlying mechanisms.

[1] Department of Psychology, University of Pennsylvania, Philadelphia, PA 19104, USA. [2] Department of Psychology, Durham University, Durham DH1 3LE, UK. [3] Department of Psychology and Logopedics, Faculty of Medicine, University of Helsinki, Helsinki 00014, Finland. [4] Department of Neurology, University of Pennsylvania, Philadelphia, PA 19104, USA. [5] Present address: Princeton Neuroscience Institute, Princeton University, Princeton, NJ 08544, USA. These authors contributed equally: Marcelo G. Mattar, Maria Olkkonen. Correspondence and requests for materials should be addressed to M.G.M. (email: marcelomattar@gmail.com)

Neural responses to visual stimuli are modulated by the preceding temporal context, a phenomenon know as adaptation[1–9]. Adaptation is often manifested as a reduction in the neural response evoked by stimuli that are identical or similar to those observed previously. This effect is observed in various brain regions and over a wide range of timescales—from milliseconds[10] to minutes[11] to days[12]—suggesting that this process is constantly at work in the nervous system[13].

Adaptation has been proposed to facilitate efficient sensory coding by tuning the response properties of neural populations to the current sensory environment[14–16]. In particular, adaptation may reduce the correlation between neural activity patterns corresponding to frequently encountered stimuli[14,17], either by shifting neuronal tuning curves away from one another, or by narrowing neuronal selectivity[16,18–21]. Empirical support for this hypothesis has been found in some animal studies: neurophysiological recordings from monkey primary visual cortex show that adaptation to stimulus orientation decorrelates neural responses[22,23], and recordings from cat primary visual cortex show that adaptation promotes population homeostasis[24]. In humans, adaptation improves fMRI decoding of numerosity in the intraparietal sulcus[25].

Adaptation also alters perception[26]. Psychophysical data in humans collected with low-level stimuli (e.g., color, motion direction, speed, orientation) demonstrate an improvement in perceptual discrimination following adaptation to stimuli with similar features[6,27–30] (although see ref. [31]). For more complex stimuli, however, results are mixed. Adaptation to faces, for example, does not reliably improve discrimination of face-related attributes[32–35].

Theoretical work has offered possible links between the decorrelation effects of neural adaptation and the improvements in perceptual discrimination that follow[20,36], but direct experimental evidence of simultaneous neural response and perceptual effects is lacking. Here, we tested the predictions of the efficient coding model in both the psychophysical performance of humans and in the corresponding multivariate patterns of BOLD fMRI response. Specifically, we tested whether brief exposure to a set of synthetic visual shapes drawn from a common prototype would enhance the discriminability of subsequently presented stimuli with similar features. The use of complex 3D shapes allowed us to measure the distributed pattern of voxel responses within higher-level visual cortex. To anticipate our results, we observe that perceptual discrimination of 3D shapes is enhanced after adaptation, and that multi-voxel fMRI patterns evoked by these stimuli are correspondingly decorrelated.

## Results

**Adaptation improves perceptual shape discriminability.** To test the effect of adaptation on discrimination of high-level visual stimuli, we created two sets of computer-generated three-dimensional shapes (Fig. 1a). Each set of shapes (shape class) was based upon a different prototype shape, and all items in a given shape class were more similar to each other than they were to items in the other shape class. We then asked if perceptual adaptation to stimuli from one of the shape classes would improve subsequent shape discrimination thresholds for items from that same class.

Twenty subjects performed multiple runs of a delayed match-to-sample task. Each of two experimental runs started with an adaptation phase during which subjects viewed a rapid, serial presentation of shapes from one of the sets for 60 s (Fig. 1b). After the adaptation period, subjects performed a series of discrimination trials. Each trial started with 4 s of top-up adaptation,

followed by the presentation of a single sample stimulus and, after a brief delay, the same sample stimulus shown simultaneously with one of the two prototype stimuli. Subjects indicated with a button press which of the two shapes displayed in the second interval matched the shape in the first interval (Fig. 1c). A match-to-sample task is appropriate for measuring discrimination thresholds as it does not require verbalizing the dimension on which the match is made; subjects merely have to pick the more similar shape, instead of having to pick the (e.g.) spikier one[37]. To generate a series of samples for each prototype on an arbitrary axis in shape space, we morphed the two prototypes with fine spacing. The similarity between the sample and the prototype stimulus shown on each trial was adjusted based on a 1-up, 3-down staircase procedure. We estimated the discrimination thresholds for each prototype from the last five reversals of the staircase and averaged the discrimination thresholds within subject across the two experimental runs. Trials in which the sample and prototype stimuli were drawn from the same shape class as the adaptors were interleaved with trials in which the sample and prototype stimuli were drawn from the other shape class.

The efficient coding hypothesis predicts a lower discrimination threshold (i.e., better performance) for stimuli from the adapted stimulus set as compared to the unadapted stimulus set. This result was found for all 20 subjects (Fig. 2a). We tested this effect in a three-way repeated measures ANOVA with adaptation condition (adapted/unadapted), prototype (A/B), and run (1/2) as factors. Adaptation improved discrimination thresholds, evidenced by a main effect of adaptation condition ($F(1, 77) = 58.5$, $p < 0.001$; Fig. 2a). We also observed a main effect of prototype stimulus on thresholds ($F(1, 77) = 14.4$, $p < 0.001$), indicating that subjects found it easier to discriminate stimuli within shape class A as compared to B (Fig. 2b). Post hoc tests confirmed that the adaptation effect was present for both shape classes: mean decrease in threshold was 4.6 morph units for prototype A (paired $t$-test: $t(19) = 4.8$, $p < 0.001$) and 6.4 units for prototype B (paired $t$-test: $t(19) = 8.8$, $p < 0.001$). Finally, there was no significant difference in the adaptation effect for the two runs ($F(1, 77) = 0.33$, $p = 0.57$; Fig. 2c).

**Adaptation improves voxel pattern discriminability.** We next asked whether adaptation enhances the discriminability of voxel patterns evoked in object-selective regions of the brain. We tested this hypothesis in 10 new subjects using BOLD fMRI to measure the pattern of evoked responses to probe stimuli from each shape class (Fig. 1d). The same procedure as in the psychophysics study was followed for the initial adaptation on each experimental run and for the top-up adaptation at the beginning of each trial. Following top-up adaptation, subjects were presented with a single probe stimulus derived either from the same shape class as the adaptors or from the other shape class. The ordering of the four probe stimuli across trials was counterbalanced, and a 10% size modulation was applied to each presentation. Subjects performed a cover task in which they indicated with a button press whether the probe was smaller or larger than the adapting stimuli in the adapting phase. Four experimental runs were conducted using each of the two shape classes as the adapting stimulus, for a total of eight fMRI runs per subject.

We focus our initial analysis upon the left lateral occipital cortex (left LO)[38], as LO is believed to be central to object shape perception and previous work has found that adaptation effects in LO for repeated presentations of different exemplars of the same shape category are most prominent in the left hemisphere[39,40]. We used the data from an independent functional localizer scan collected for each subject to identify the 100 voxels on the cortical

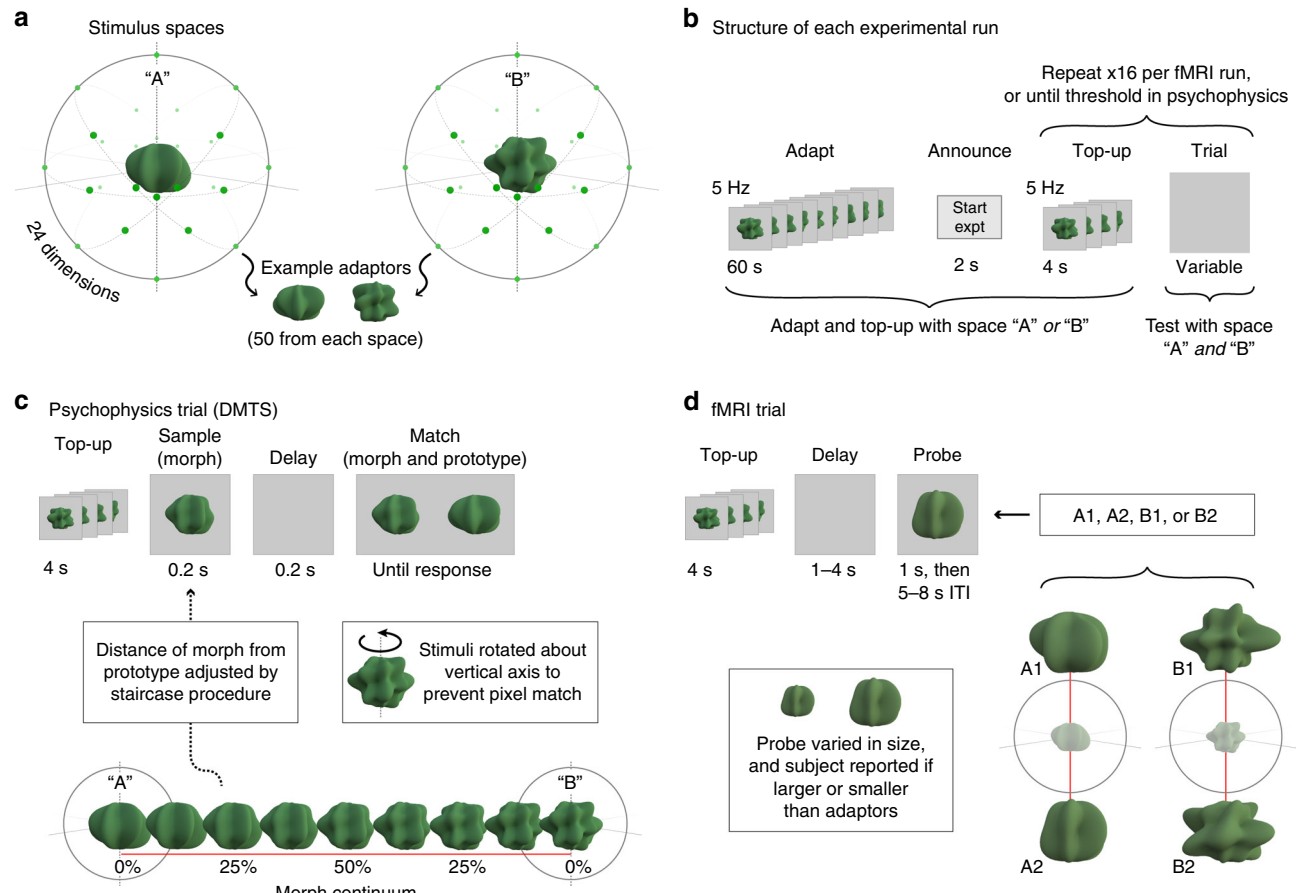

**Fig. 1** Experimental design. **a** We sampled 3D shapes from a 24-dimensional, radial frequency stimulus space in which the dimensions were defined by the frequencies, amplitudes, and phases of 8 sinusoids. Two visually distinct shapes from this space were chosen as prototypes A and B. For each prototype, 50 adaptor stimuli were generated in the multidimensional space by jittering the orientation and amplitude of the 8 sinusoidal components, producing two distinct shape classes. **b** In the beginning of each experimental run (for both psychophysics and fMRI experiments), subjects were presented with a series of shapes from one of the two shape classes for 60 s at a rate of 5 Hz (adapt phase). All trials began with a top-up adaptation period of 4 s at 5 Hz, followed by trial events that differed for the psychophysics and fMRI experiments (**c, d**). In all experiments, adaptation condition (A or B) was blocked in counterbalanced order across subjects. **c** Morph stimuli were generated along the direction that connected the two prototype shapes. Subjects first saw a morph and, after a brief delay, the same morph and a prototype. Subjects were then asked to report which of the two shapes in the second interval matched the first shape (i.e., which of the two shapes was a morph). To avoid pixel-based matches, each morph and prototype shape was displayed as a static image in one of two possible rotations. Trials using A and B as prototypes were interleaved in a run, and separate 1-up, 3-down staircases controlled the morph value for each prototype. **d** Two distinct probe stimuli were created for each shape class (A1, A2 and B1, B2) along axes passing through the prototype stimuli. After a top-up adaptation phase and a variable interstimulus interval, subjects saw one of the four probe stimuli and were asked to report whether the probe was larger or smaller than the adaptors in the top-up phase. We measured the BOLD response to each of the four probe stimuli and calculated the multivariate pattern similarity between the two probe shapes (e.g., A1 and A2) after adaptation to the same shape class (A) compared to adaptation to the different shape class (B)

surface within a left LO parcel that had the greatest differential response to shapes vs. scrambled shapes. In each subject, we measured the average amplitude of evoked response for each of the four probe stimuli at each of the 100 voxels, and then obtained the pairwise, Pearson correlation between the patterns of response to each of the four probes (Fig. 3a).

The efficient coding hypothesis predicts that neural representations of adapted stimuli become less correlated with one another, in keeping with an increased ability to discriminate their identity. In our experiment, this effect would be manifest as a decrease in the correlation between the voxel responses evoked by a pair of probe stimuli (e.g., A1 and A2) when those stimuli are preceded by a matching adapting phase. We find evidence for this effect in our data (Fig. 3b). Using a repeated measures ANOVA, we examine the influence of adaptation condition (adapted/unadapted) and prototype (A/B) on the correlation

between the voxel responses to the two probe shapes. We observed that the main effect of adaptation condition was significant ($F(1, 9) = 25.62$, $p = 0.0007$), indicating that pairwise similarity after adapting to the same shape class was lower than after adapting to the other shape class. Neither the main effect of prototype ($F(1,9) = 0.0096$, $p = 0.92$) nor the interaction term were significant ($F(1, 9) = 3.42$, $p = 0.098$), suggesting that the decorrelation effect was not significantly different between the two prototypes. However, in post hoc tests examining this effect in the two stimulus spaces separately, the correlation between the B probe stimuli (B1 and B2) was lower when subjects were adapted to shape class B than when they were adapted to shape class A (paired $t$-test on Fisher $z$-transformed correlation coefficients: $t(9) = 4.83$, $p = 0.0009$), but the complementary test with the A shape class was not significant (paired $t$-test on Fisher $z$-transformed correlation coefficients: $t(9) = 1.26$, $p = 0.24$;

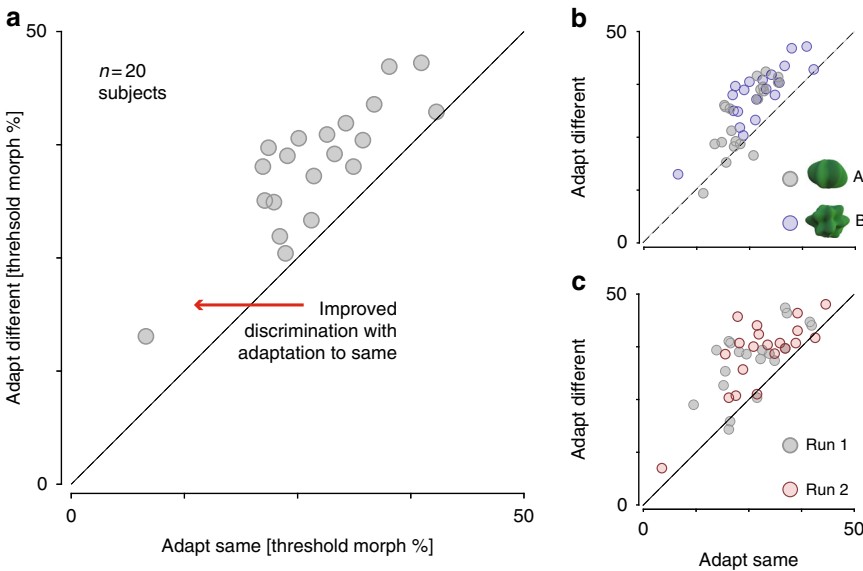

**Fig. 2** Perceptual shape discriminability. **a** Shape discrimination thresholds are plotted for each subject ($n = 20$) after adapting and testing for the same shape class (x-axis) vs. adapting and testing to different shape classes (y-axis). Each data point is the average over two repetitions and over both prototype shapes. All data points lie above the diagonal, indicating that shape discrimination thresholds were lower after adapting to the same shape class for every subject. **b** Thresholds shown separately for the two prototype shapes, averaged over runs. The two distributions are partially non-overlapping, suggesting that shape class A was easier to discriminate overall than shape class B. **c** Thresholds shown separately for the two runs, averaged over shape

Fig. 3b, c). We note that the stronger decorrelation observed for shape class B is consistent with the larger decrease in perceptual discrimination thresholds for these stimuli reported above.

We also examined the consistency of the decorrelation effect across subjects. For each subject, we measured the Pearson correlation of the voxel responses evoked by a pair of probe stimuli when those stimuli were from the same class as the adaptors, and when they were from the unadapted shape class (Fig. 3d). All ten subjects had a lower correlation between probe stimuli in the adapted condition (paired t-test on Fisher z-transformed correlation coefficients: $t(9) = 5.06$, $p < 0.0007$). Examining the effect in the two stimulus spaces separately we found that 6 out of 10 subjects had a lower correlation between A probe stimuli in the adapted condition, and that 9 out of 10 subjects had a lower correlation between B probe stimuli in the adapted condition (Fig. 3e).

We then examined the cortical extent of these effects. We performed a searchlight analysis[41], calculating within each volumetric searchlight (5 mm radius) the magnitude of the pattern decorrelation induced by adaptation (Fig. 3f). A broad swath of posterior occipital–temporal cortex evidenced pattern decorrelation, with the largest effect sizes observed within the occipital pole, lateral occipital cortex, and the posterior region of fusiform gyrus. The effect was significant in clusters within the occipital and lateral occipital cortices, as well as a cluster in the temporal lobe (Fig. 3g). Thus, our findings generalize beyond the left LO ROI used in the previous analyses.

Taken together, the behavioral and fMRI results support the efficient coding hypothesis: adaptation to a given shape class produces both a behavioral improvement in the discriminability of similar shapes, as well as an increased separability of their multi-voxel patterns in object-selective cortical regions and nearby visual areas.

**Mechanisms of decorrelation.** As adaptation is known to reduce the amplitude of BOLD response, one possible mechanism for our findings is a reduction in stimulus-evoked responses. If this

reduction in response occurs in the setting of independent, unchanged measurement noise, the reduced correlation we observe in the patterns evoked by adapted stimuli may be the product only of a lower signal-to-noise ratio. Consistent with this mechanism, we find that the evoked BOLD fMRI signal amplitude is smaller in the adapted as compared to the unadapted condition within left LO (percent signal change in adapted condition: $1.30 \pm 0.16\%$ (SEM across subjects, $n = 10$) vs. unadapted: $1.45 \pm 0.16\%$; paired t-test: $t(9) = 2.78$, $p = 0.0213$). A similar response reduction was observed for either shape class (percent signal change for shape class A in adapted condition: $1.28 \pm 0.15\%$ vs. unadapted: $1.38 \pm 0.15\%$; percent signal change for shape class B in adapted condition: $1.33 \pm 0.19\%$ vs. unadapted: $1.51 \pm 0.21\%$).

We find, however, that this reduction in amplitude varies markedly across voxels. We calculated the magnitude of response suppression for each voxel as a scaling factor between adapted and unadapted responses, and then identified sets of voxels with different degrees of response suppression. The 50 voxels with lowest values for this index had a mean suppression value of $0.82 \pm 0.05$ (i.e., the BOLD fMRI signal evoked by the stimuli was reduced on average by 18% in the adapted condition). In contrast, the 50 voxels with the largest values for this index actually demonstrated response enhancement in the adaptation condition ($1.17 \pm 0.08$, or an increase by 17% of response amplitude in the adaptation condition). If a reduction in response amplitude alone accounts for the decorrelation of patterns that we observe in the adaptation condition, then a decorrelation effect should not be present in the subset of voxels with response enhancement. In disagreement with this account, a significant decorrelation effect was still found (paired t-test on Fisher z-transformed correlation coefficients: $t(9) = 2.50$, $p = 0.034$), although decorrelation was marginally stronger for the subset of voxels with most suppression (paired t-test on Fisher z-transformed correlation coefficients: $t(9) = 2.2049$, $p = 0.055$). To analyze more closely the relationship between the suppressive effect of adaptation and the decorrelation effect, we grouped voxels into various bins according to the degree of response suppression (scaling factor)

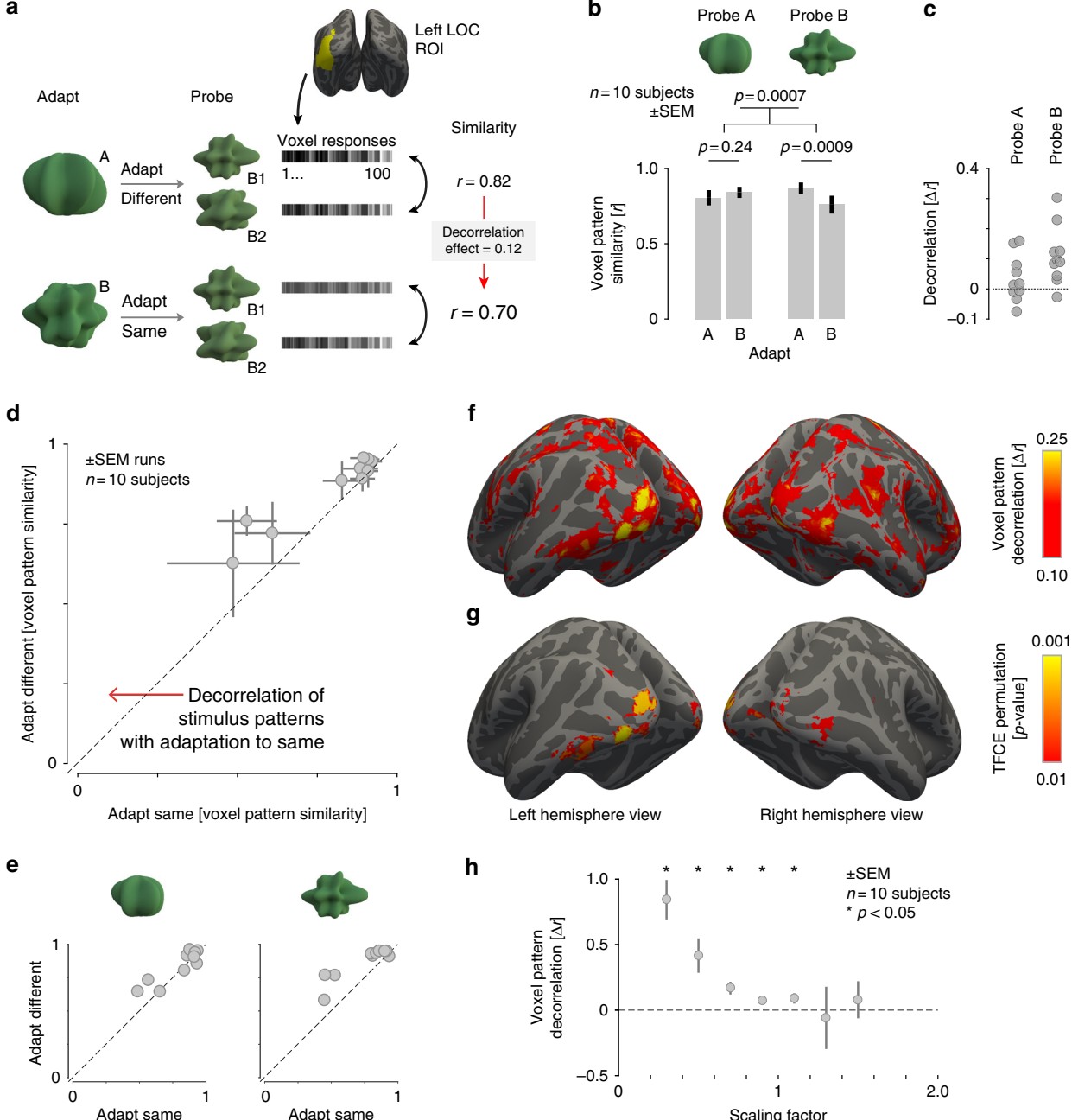

**Fig. 3** Voxel pattern discriminability. **a** Schematic of the fMRI decorrelation analysis. We measured responses in left LO to a pair of probe shapes after adaptation to either shape class A or B. We then computed the similarity of voxel responses evoked by the pairs of probe shapes, and compared the resulting Pearson's correlation for the probe shapes that were either adapted or unadapted. A decorrelation effect was defined as the difference between these two correlation values. The logic of the analysis is illustrated for the test in which shape class B served as the probe. **b** Voxel pattern similarity for probe stimuli from the A (left) and B (right) shape classes. Patterns were more distinct (less correlated) for probe shape pairs after undergoing adaptation to the same shape class. Error bars are standard error of the mean across subjects ($n = 10$). $p$-values are shown for paired $t$-tests. **c** Magnitude of the decorrelation effect for individual subjects. **d** Correlation between multi-voxel patterns for each subject, comparing the condition in which probe stimuli were from the same space as the adaptors ($x$-axis), to when they were from the different space ($y$-axis). For every subject, patterns from adapted pairs of stimuli were less correlated than patterns from unadapted stimuli. Each data point represents one subject and error bars represent variability (SEM) across scan acquisitions within subject. **e** Correlation between multi-voxel patterns for each subject displayed separately for each probe. **f** A whole-brain searchlight analysis presenting the mean, across-subject magnitude of the decorrelation effect across the cortex, thresholded at $\Delta r > 0.10$. **g** A whole-brain decorrelation significance map obtained from a permutation test with threshold-free cluster enhancement, thresholded at $p < 0.01$. **h** Voxel pattern decorrelation measured separately for groups of voxels with varying degrees of response suppression. The scaling factor of each voxel was measured as the ratio between mean response in the adapted condition vs. mean response in the unadapted condition. Error bars represent variability (SEM) across subjects and asterisks denote significant ($p < 0.05$) decorrelation assessed with a paired $t$-test

exhibited within a scan. We then computed the degree of pattern decorrelation for each bin containing 10 or more voxels. We found that decorrelation effect is largest in voxels whose responses are most suppressed, suggesting that this effect is directly or indirectly linked to the ubiquitous reduction in the amplitude of evoked responses (Fig. 3h), although response suppression alone cannot account for the observed pattern decorrelation effects.

## Discussion

We used behavioral and neuroimaging data to test the hypothesis that adaptation to high-level stimulus features decorrelates stimulus representations. In a psychophysical experiment, we found that adaptation improves the perceptual discriminability of similar shapes. In a BOLD fMRI experiment, we found that adaptation enhanced the distinctiveness of voxel responses to perceptually similar stimuli. While the fMRI effect was concentrated in object-selective cortex, a searchlight analysis demonstrated similar results in nearby regions of the occipital and ventral temporal lobes. Our results offer three novel contributions: (i) we provide evidence for enhanced perceptual discriminability of high-level stimuli (3D shapes) following adaptation, thus clarifying earlier findings whose results were mixed[32–35] and building on previous findings from a different domain[25]; (ii) we demonstrate a decorrelation of voxel pattern representations in human observers undergoing visual adaptation; (iii) we offer joint behavioral and neuroimaging evidence using a similar experimental paradigm, thus offering a link between the neural effect and its behavioral consequences.

Our analyses provide some preliminary insight into the mechanism of the voxel pattern decorrelation effect. Because decorrelation was present even in the absence of response suppression, a uniform reduction in evoked response (in the face of an unchanged level of measurement noise) cannot completely account for our results. Decorrelation could instead be driven by a non-homogenous scaling of responses, such as a larger response reduction in weakly responsive units. Such an effect could serve to maintain population homeostasis in sensory cortices[24], and has found empirical support as a sharpening effect in fMRI studies of adaptation[42,43]. Computationally, a sharpening of neural tuning curves has been demonstrated to produce both perceptual biases and a decrease in perceptual discrimination thresholds[20,21]. In our data, a larger decorrelation effect was observed for voxels undergoing most response reduction, suggesting a link between these two effects of temporal context.

While our study demonstrates perceptual improvements in discrimination performance and voxel changes in representation using the same adaptation procedure, we note that our findings do not directly relate these phenomena. An ideal model would provide a quantitative mapping between neural and perceptual effects on a trial-by-trial basis. A challenge to such an effort is that measurement of the voxel responses is complicated in the face of a perceptual task that requires the subject to explicitly process the similarity of presented stimuli, since the behavioral task could produce confounding effects in the neural data. Alternatively, if decorrelation is a stable property of the individual and with sufficient inter-subject variability, a link could be established by measuring both perceptual and neural effects on the same individuals and examining whether the effects co-vary. A complete model would also account for the cortical extent over which this decorrelation effect is observed. We find that visual cortex broadly demonstrates the decorrelation effect, although it is of greater strength in object-responsive areas that have been previously shown to exhibit coarse spatial coding for object shape[44].

Our paradigm bears some resemblance to category learning, especially as discussed in prototype theory (e.g., ref. [45]), though it differs in two important ways. First, instead of learning to discriminate A from B, our subjects had to discriminate within category A or B, which is a question not typically addressed in category learning studies (although see ref. [46]). Second, our study examines how discrimination is affected by the preceding few seconds and minutes of exposure, a timescale shorter than often considered in category-learning studies. Our study also shares similarities with perceptual learning paradigms, in which subjects learn to discriminate between two initially indistinguishable stimuli after several hours or days of practice[47]. While the phenomena of category and perceptual learning manifest over much longer time scales than the seconds-to-minutes of adaptation studied here, we note that adaptation and learning mechanisms over longer timescales may share underlying neural mechanisms[13], in line with observations of repetition suppression over intervals as long as multiple days[12].

In summary, we find evidence in support of adaptation as a mechanism for representational efficiency. In line with theoretical predictions, decorrelated representations in the brain accompany an improvement in perceptual discrimination.

## Methods

**Subjects**. For Experiment 1 (psychophysics), 20 healthy human subjects (15 females, ages 18–51 years) with normal or corrected-to-normal vision were recruited through the University of Pennsylvania subject pool. Subjects were compensated $10/h, and informed consent was obtained from all subjects. The experimental protocol was approved by the University of Pennsylvania Institutional Review Board.

For Experiment 2 (neuroimaging), 10 healthy human subjects (6 females, ages 21–37 years) with normal or corrected-to-normal vision were recruited through the University of Pennsylvania subject pool. Subjects were compensated $20/h, and informed consent was obtained from all subjects. The experimental protocol was approved by the University of Pennsylvania Institutional Review Board. None of the 10 subjects participated in both psychophysics and neuroimaging experiments.

**Psychophysics stimuli and protocol**. To measure shape discrimination thresholds, we constructed a parametric shape space with ShapeToolbox[48] (URL: http://github.com/saarela/ShapeToolbox). ShapeToolbox generates three-dimensional, radial frequency patterns by modulating basis shapes such as spheres with an arbitrary combination of sinusoidal modulations in different frequencies, phases, amplitudes, and orientations. Specifically, we generated shapes by modulating a sphere with different combinations of six different sinusoids. Each sinusoid had four parameters: component frequency, amplitude, phase, and orientation, resulting in a 24-dimensional shape space. We then chose two shapes as prototype shapes, which were sufficiently different that they were intuitively perceived as members of two different shape categories (Fig. 1a). The same two prototype shapes (prototype shape A and prototype shape B) were used for all subjects.

Subjects viewed the stimuli in a dim room from a distance of ~57 cm on a Macbook pro laptop (15″ retina display, resolution 2889 × 1800). Discrimination thresholds for the two prototype shapes were measured in interleaved trials embedded in three different types of run. In the baseline runs, discrimination thresholds were measured without adaptation to shape. In the adapt A runs, a rapid stream of adaptors that were similar to prototype shape A were shown in the beginning of the run for 60 s, and correspondingly for adapt B runs. In the adapting phase, each adaptor was presented for 150 ms with a 50 ms ISI, or at a rate of 5 Hz (Fig. 1b). The adaptors for each prototype shape were created by jittering the orientation and amplitude of the sinusoidal modulation of the particular prototype while maintaining the Euclidean distances in the multidimensional shape space between the prototype and the adaptors, thus creating a region in shape space around the prototype shape (Fig. 1a). The variability in parameter space of these adaptors (50 total) was held constant and smaller than the variability between the two stimulus categories. To ensure focus on the stimuli during the adaptation periods, a fixation cross in the middle of the display randomly changed color from black to gray, and subjects were asked to indicate the color changes with a key press. The end of the adaptation period was indicated to the subject with the text start experiment. Subjects first practiced the baseline discrimination task, after which they ran one baseline run. They then practiced the adaptation task, and ran each adaptation run twice in ABAB or BABA order (counterbalanced across subjects). We collected the baseline data originally to allow us to quantify the effect of adaptation in relation to baseline discrimination. We however converged on an analysis for both psychophysics and fMRI where we compared the thresholds for

the two shape classes as a function of adaptor (same vs. different class). Thus, we do not report the baseline data here.

Shape discrimination was measured with a delayed match-to-sample procedure. On each trial, subjects saw a test stimulus in the center of the screen for 200 ms, and then after a delay of 250 ms the same test stimulus was paired with either prototype A or B and displayed until response. Subjects were asked to indicate which of the two shapes in the second interval best matched the first shape. To avoid pixel matches, the tests and the prototype were shown in one of two possible vertical rotations. This ensured that the subject had to extract the 3D shape of the stimuli to make the match.

Morph stimuli were generated by morphing the two prototype shapes at equal intervals along an 80-step morph continuum (Fig. 1c). The test shape on each trial was selected according to an adaptive staircase procedure that was separately run for the two prototypes, with stimuli in each case drawn from the 40 steps closest to the prototype. Each of the two staircases converged on the 79% discrimination threshold (1-up, 3-down rule). The staircases ended after eight reversals. The discrimination threshold was taken as the average of the last five reversals. Each trial was preceded by a top-up adaptation period with stimuli presented for 4 s at 5 Hz. To characterize the effect of adaptation on discrimination, we compared thresholds for each prototype shape after adapting to spaces A and B.

**Neuroimaging stimuli and protocol**. The same prototype shapes and pool of adaptors from the behavioral experiment were used in the fMRI experiment. We additionally generated two probe stimuli for each shape class as follows. For each prototype, we defined a random direction in our parametric shape space and selected two shapes along this dimension, equally distant from the prototype in the positive and negative direction (Fig. 1d). While the random direction was not constrained to be orthogonal to the shape space vector that connects the two prototype shapes, the high-dimensionality of the shape space essentially guaranteed this property. The same probe stimuli were used for all subjects and across all runs.

Each experimental run (corresponding to one scan acquisition) began with a 60 s pre-adaptation phase during which a stream of adaptors from one of the shape class was presented at a rate of 5 Hz in random order. Subjects then completed 16 trials, each starting with a top-up adaptation phase for 4 s at the same 5 Hz rate, followed by a jittered ISI (1–4 s) and a test stimulus selected from the four possible probes (A1, A2, B1, B2), presented for 1 s. We used a cover task on each trial to ensure attention to the stimuli. A size modulation of 10% (larger or smaller) was applied to the probe stimuli, and subjects were instructed to select with a left/right button press whether the stimulus was smaller/larger than the stimuli in the top-up phase. Trial onsets were separated by 14 s, so the interval between the offset of the probe stimulus and the onset of the top-up phase of the next trial was 5–8 s. The size modulation of each shape and the order of the test stimuli within each run were counterbalanced (four instances of each test stimulus per run, each presented twice in smaller/larger versions). Subjects completed a total of eight scans, four with each shape class used in the adaptation phases, in counterbalanced order. Stimuli occupied 10° of visual angle (with a size modulation of 10% for test stimuli) and were presented on a gray background.

In addition to the main experimental scans, subjects completed one or two functional localizer scans consisting of 16-second blocks of faces, objects, scenes, and scrambled objects. Images were presented for 800 ms with a 200 ms ISI, and subjects performed a one-back task on image repetition.

**Neuroimaging data acquisition and preprocessing**. Magnetic resonance images were obtained at the Hospital of the University of Pennsylvania using a 3.0 T Siemens Trio MRI scanner equipped with a 32-channel head coil. Stimuli were displayed on a Sanyo-PLC-XT35 LCD projector and viewed via a mirror mounted on the head coil. The viewing area of the display was $50.5 \times 38$ cm or $23 \times 17°$ of visual angle. Stimulus size was $\sim 10 \times 10°$ of visual angle. T1-weighted structural images of the whole brain were acquired on the first scan session using a three-dimensional magnetization-prepared rapid acquisition gradient echo pulse sequence (repetition time (TR) 1620 ms; echo time (TE) 3.09 ms; inversion time 950 ms; voxel size $1 \times 1 \times 1$ mm; matrix size $190 \times 263 \times 165$). A field map was also acquired at each scan session (TR 1200 ms; TE1 4.06 ms; TE2 6.52 ms; flip angle 60° voxel size $3.4 \times 3.4 \times 4.0$ mm; field of view 220 mm; matrix size $64 \times 64 \times 52$) to correct geometric distortion caused by magnetic field inhomogeneity. Functional data were acquired with T2*-weighted images sensitive to blood oxygenation level-dependent contrasts using a slice accelerated multiband echo planar pulse sequence (TR 2000 ms; TE 25 ms; flip angle 60°; voxel size $1.5 \times 1.5 \times 1.5$ mm; field of view 192 mm; matrix size $128 \times 128 \times 80$, acceleration factor 4). 105 volumes were acquired for the functional localizer(s) scans, and 142 volumes were acquired for each experimental scan.

Cortical reconstruction and volumetric segmentation of the structural data was performed with the Freesurfer image analysis suite[49]. Boundary-Based Registration between structural and mean functional image was performed with Freesurfer bbregister[50]. Preprocessing of the fMRI data was carried out using FEAT (FMRI Expert Analysis Tool) Version 6.00, part of FSL (FMRIB's Software Library, www. fmrib.ox.ac.uk/fsl). The following pre-statistics processing was applied: EPI

distortion correction using FUGUE[51]; motion correction using MCFLIRT[52]; slice-timing correction using Fourier-space time series phase-shifting; brain extraction using BET[53]; grand-mean intensity normalization of the entire 4D dataset by a single multiplicative factor; highpass temporal filtering (Gaussian-weighted least-squares straight line fitting, with $\sigma = 50.0$ s). Spatial smoothing was performed with a 5 mm FWHM kernel using FSL's `fslmaths`.

**General Linear Model**. Statistical analyses were performed upon the time-series data from each functional run using a General Linear Model in FSL. Analysis of the main experimental scans included one binary covariate for the pre-adaptation phase, one binary covariate for all 16 top-up phases, and binary covariates for each of the four test stimuli, each convolved with a canonical double-gamma hemo-dynamic response function. The temporal derivatives of each covariate were also included. The resulting statistical maps were projected to surface space using Freesurfer's `mri_vol2surf`.

Analysis of the functional localizer scans included binary covariates for the four stimulus types (faces, objects, scenes, and scrambled objects) and a binary covariate for button-press. Parameter estimates were averaged across scans for those subjects that completed two functional localizers. The resulting statistical maps were projected to surface space using Freesurfer's `mri_vol2surf`.

**ROI definition**. We focused our analyses on a well-characterized object-selective area of the human brain, the Lateral Occipital Complex LOC[38]. The human LOC is divided into a ventral part in the posterior fusiform gyrus (pFus) and a region in the lateral occipital cortex (LO). Previous studies suggest the existence of a coarse spatial coding of shape features in LO and a more focused coding of the entire shape space within pFus[44]. Studies focusing on laterality differences in LOC have reported larger adaptation effects for repeated presentations of different exemplars of the same object category in the left hemisphere[39,40], and greater sensitivity to changes in viewpoint in the right hemisphere[54]. We therefore focused on the left lateral occipital cortex (left LO) as our region of interest due to its greater sensitivity to shape similarity at the multi-voxel scale, and greater invariance to category and size information.

We defined left LO for each subject as the 100 voxels on the cortical surface that responded most strongly in the functional localizer scan to the contrast of objects vs. scrambled objects within a larger group-defined left LO parcel warped to the subject's own surface space (Fig. 3a). This method ensures sensitivity to the between-subject variability of the spatial location of this ROI[55].

**Multi-voxel pattern analyses and searchlight**. ROI pattern analysis was performed on the parameter estimates (beta values) within the left LO extracted from the GLM. Whole-brain pattern analyses were performed using a volumetric searchlight procedure[41], with analyses performed on the parameter estimates within every 5 mm sphere in the brain. In both analyses, we considered separately the multi-voxel patterns evoked by adapted shapes (when the test stimulus was from the same space as the adaptors) and the multi-voxel patterns evoked by unadapted shapes (when the test stimulus was from the other, unadapted space). For each ROI or searchlight, the correlation between the two patterns evoked by the adapted shapes and the correlation between the two patterns evoked by the unadapted shapes were calculated for each experimental run, their Fisher $z$-transformations averaged across scans, and the averaged values compared (Fig. 3a). We assessed significance at the whole-brain level using threshold-free cluster enhancement (TFCE[56]), an algorithm designed to offer the sensitivity of cluster-based thresholding without the need to set an arbitrary threshold. We corrected the TFCE map for familywise error rate using FSL's 1-sample group-mean permutation test (exhaustively testing all 1024 permutations) and spatial 10 mm FWHM variance smoothing to reduce noise from poorly estimated variances in the permutation test procedure. Searchlight results are presented on the surface (Freesurfer's `fsaverage`) both with an effect size map (using a threshold of $\Delta r = 0.10$; Fig. 3e) and with a significance map (using a threshold of $p = 0.01$; Fig. 3f).

**Mechanisms of decorrelation**. We calculated the average amplitude of evoked response for each voxel in both adapted and unadapted states, averaging the patterns evoked by each stimulus across runs. The degree of response suppression for each voxel was defined as the ratio between the average response for adapted shapes and the average response for unadapted shapes. We use this index to group voxels into various bins from which we computed pattern decorrelation. Only bins containing 10 or more voxels and at least 7 subjects were included in this analysis.

**Code availability**. All analyses were conducted using custom code written in MATLAB v9.1.0 (R2016b), which is available from the corresponding author upon reasonable request.

## Data availability

The data that support the findings of this study are available from the corresponding author upon reasonable request.

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

## Acknowledgements

This project was supported by NIH R21-EY-022751-02, Core Grant for Vision Research P30 EY001583, and the Neuroscience Neuroimaging Center Core Grant P30 NS045839. MO received support from the Academy of Finland Academy Fellow program, grant 287506. We thank Toni Saarela for valuable help with stimulus generation, and for the staircase code and Toni Saarela and Stacey Aston for helpful comments on the manuscript.

## Author contributions

Conceptualization: M.G.M., M.O., R.A.E. and G.K.A.; Investigation (fMRI): M.G.M.; Investigation (psychophysics): M.O.; Formal Analysis: M.G.M. and M.O.; Supervision:

R.A.E. and G.K.A.; Writing—Original Draft: M.G.M. and M.O.; Writing—Review & Editing: M.G.M., M.O., R.A.E. and G.K.A.; Funding Acquisition: R.A.E.

## Additional information

**Competing interests:** The authors declare no competing interests.

