## [Peer Review File · Nature Communications]

Reviewers' comments:

Reviewer #1 (Remarks to the Author):

The manuscript reports a study that uses fMRI to test the hypothesis that adaptation results in neural decorrelation and enhanced perceptual discriminability of similar stimuli. The study employs a highly controlled object space that allows the investigators to test this question in higher-level visual areas. The behavioral results are convincing. However, the fMRI findings are less conclusive.

In particular, the fMRI effect of decorrelation was only shown significantly in space B but not space A. Further methodological details are needed to assess how the voxel pattern similarity was estimated. Figure 3c- the key figure showing the effect of decorrelation is not convincing: most points are on the diagonal with error bars crossing the diagonal rather than shifted towards the y axis.

The searchlight analysis shows patterns extending in the visual and parietal cortex; however, there is no mention of control analysis (e.g. permutation tests, comparison with shuffled labels) or significance testing (was there an arbitrary threshold chosen for the maps in Figure 3d?).

Further, the behavioral and fMRI experiments were conducted on different participants. As a result, linking behavioral and fMRI findings is not possible. This compromises the link between perceptual discriminability and neural decorrelation due to adaptation. The behavioral effect was stronger for stimuli in space A rather than B. However, the fMRI study did not show a significant effect for space A. How can these findings be reconciled?

An attentional control task was performed in the scanner. However, performance in this task was not reported. Therefore, it is difficult to assess whether attention was controlled for or whether differences in attention across conditions may drive stronger decorrelation of the fMRI signals. The searchlight analysis revealed parietal regions known to be involved in attentional processing. Was decorrelation in these regions due to attentional confounds? Stronger evidence is needed to discern whether neural decorrelation was due to adaptation rather than differential attention.

Two sets of vertices are separated based on univariate response. Both these clusters are shown to relate to decorrelation. However, no statistical comparison is reported that tests which of the two clusters shows stronger decorrelation. This is important for assessing whether the lower signal to noise ratio accounts for reduced correlation. Further, the two clusters may have different variance, but this is not taken into account when comparing these voxels; yet difference in the variance may affect the decorrelation analysis.

An attempt is made to compare the results with other models of adaptation: sharpening vs. fatigue. Again here two sets of vertices are compared; however the statistics are not convincing (why is 1-tail t-test used?). Further, previous imaging papers have used multivariate analysis and compared voxel-tuning curves to provide evidence for sharpening. Further analysis of the data following this methodology should be conducted to conclusively infer whether decorrelation relates to sharpening rather than fatigue. The data currently presented remain unconvincing in discerning between the neural mechanisms that underlie adaptation.

Reviewer #2 (Remarks to the Author):

Mattar and colleagues present a careful and well-conducted study investigating the impact of prior exposure on both behavioral shape judgments and BOLD fMRI responses in shape-selective cortical

regions. This issue has been much debated in recent years and major strengths of this study are the combination of behavioral and fMRI experiments and the elegant and simple design that precisely targets the potential role of decorrelated neural responses.

However, I think the manuscript can be strengthened by expanding the discussion of relevant literature and extending the presentation of some of the results. The sample size for the fMRI experiment is relatively small ($n=10$), but the authors show the results at an individual subject level (e.g. Figure 3C) and I don't think this is a major concern.

Major Suggestions:

- 1) The authors use the terminology of 'adaptation', but the impact of prior exposure on neural responses has also been discussed in terms of 'repetition suppression' (particularly in the neuroimaging field) and even 'habituation'. There is much literature trying to relate repetition suppression of the BOLD signal to behavior and while the authors paradigm assumes a short time frame for 'adaptation', repetition suppression effects have been shown to last longer than 24 hours (e.g. work by Alex Martin and colleagues). I think the manuscript would benefit from a broader discussion of this prior literature, placing the current results more in context and broadening the current focus (particularly in the introduction) from primarily neurophysiological studies (especially given that the current study uses fMRI).
- 2) The authors found a difference in thresholds between the two stimulus spaces (easier discrimination for A), a difference in the decrease of behavioral thresholds (greater decrease for B), and no difference in correlation of BOLD responses for stimulus space A when adapted/unadapted. However, I could find no discussion of how these results might relate to each other - I think such a discussion would be worth adding.
- 3) Given the different results for sets A and B in the fMRI data, I think the authors should show the consistency of the effects across subjects for each set separately.
- 4) Similarly, I think the authors should report the BOLD amplitudes separately for sets A and B.
- 5) The analysis in which the authors selected the vertices showing a response enhancement is very nice, but it would be helpful to compute the decorrelation effect for the vertices showing the strongest response reduction as well for comparison (i.e. is there any difference in the decorrelation effect between these sets of vertices).
- 6) It would be interesting to know how the correlation between sets A and B (and not just within set) is affected by adaptation.

Minor Comments:

- 1) Given that the authors are collecting BOLD fMRI data, I think they should be more cautious in their use of 'neural'. For example, at the start of the discussion, I don't think it is appropriate to say, 'We used....neural data...'. The same concern applies elsewhere in the manuscript.
- 2) The authors point out that trial-by-trial data would be nice, but the current data is limited not just because such trial-by-trial analyses are not possible, but more simply because the behavioral and BOLD data were not collected simultaneously and were even collected in different subjects.
- 3) The authors briefly mention category learning studies and suggest that in such studies the discrimination is typically between- and not within-category as in the current study. However, I think the study by Op de Beeck and colleagues (2006, J. Neurosci) involved within-category learning with an analytical focus on the correlation of responses to the different categories.

Reviewer #3 (Remarks to the Author):

Summary

The paper presents a very nice hypothesis-driven use of MVPA with a carefully-constructed stimulus space, to test the computational hypothesis that one of the effects of adaptation is to decorrelate neural responses to stimuli similar to the adaptor. The paper was clear, succinct, and enjoyable to read.

Two experiments are performed, in different groups of observers, one psychophysical and one using fMRI. Both use shape stimuli drawn from a 24-dimensional generative shape space. Adapting stimuli are drawn from the regions closely surrounding one of two arbitrary "prototype" points in the space. In the psychophysical experiment, perceptual shape aftereffects are quantified by measuring discrimination thresholds in a delayed 2AFC match to sample task along a 1D trajectory in the shape space connecting the two prototype shapes, after adapting to one or the other prototype regions. A similar protocol is used in the fMRI experiment, in which participants adapt to one of the two prototype regions (blocked across runs) and are shown one of four probe stimuli, two each drawn from the areas near the two prototypes (probe stimuli randomly interleaved within each run). The Pearson correlation between BOLD patterns evoked by pairs of stimuli drawn from the adapted vs unadapted regions in shape space are then calculated and compared. The main result of the paper is that shapes from an adapted region of shape space evoke less correlated patterns than do shapes from an unadapted region, which is consistent with the possibility that responses of individual neurons become less correlated following adaptation.

Major comments

1. The evidence for a causal link between the observed neural pattern decorrelation and behavioural adaption aftereffect could have been substantially strengthened by showing a positive relationship between the degree of pattern decorrelation and perceptual aftereffect in the same individuals. The variability in both of these effects was reasonably large across individuals, but unfortunately the same participants were not used in the fMRI and psychophysical experiments so this test cannot be done.
2. There is some similar previous work which is not discussed. In particular, I am aware of Castaldi et al. (2016) "Effects of adaptation on numerosity decoding in the human brain" *NeuroImage* (<https://www.ncbi.nlm.nih.gov/pubmed/27622396>). That study presents dot displays of various numerosities and uses a linear support vector machine to classify numerosity from fMRI-measured cortical patterns. They find that reasonable classification is achievable both before and after adaptation, but that adaptation improves decodability (consistent with the report here of decreased correlation between patterns after adaptation). They also test the brain-behaviour link described in the point above, showing that improvement in decodability after adaptation is correlated across participants with the behavioural numerosity aftereffect. Since that study examines a different modality and brain region of interest, and uses the different-but-related metric of SVM-classifiability rather than correlation, it does not undermine the novelty of the results presented here. However, it does reduce the apparent methodological innovation, and deserves mention in the Introduction.
3. The sharpening vs fatigue analysis on page 10 does not seem to be a good test of those hypotheses. The prediction of the sharpening hypothesis is that neurons that respond more weakly to begin with to the adapting stimuli are most suppressed by adaptation. But here, we don't have measures of the patterns elicited by the adapting stimuli before and after adaptation. Instead (if I understand correctly) vertices are categorised according to their response after adaptation, to either adapted or non-adapted probe shapes. To consider a simpler analogy: imagine we measure V1 single-cell responses to vertical and 45 degree gratings after adapting to vertical gratings, and find that firing rates are on average lower to vertical test gratings than 45 degree ones. We now categorise the cells

based on their response (after adapting to vertical) to the unadapted 45 degree gratings. We find that the cells that respond most weakly to 45 degree gratings after vertical adaptation are those most dramatically low in activity when shown vertical gratings. In this case, it seems clear that this result cannot be interpreted either for or against the sharpening hypothesis; the neurons responding weakly to the unadapted stimulus may also have had a low response to the adapting stimulus prior to adaptation (e.g. because they prefer horizontal orientations) in which case their suppressed vertical response would be consistent with sharpening, or they may have had high responses to the adaptor prior to adaptation (e.g. they are tuned to vertical) in which case the results are simply demonstrating adaptation. In other words, because baseline responsiveness is unknown, the analysis seems to confound stimulus preference with the effect of adaptation in difficult to interpret ways. It only makes sense if we assume that those neurons which respond weakly to the non-adapted stimulus also responded weakly to the adapted stimulus prior to adaptation. Finally, it also seems questionable to assume that differences in estimated BOLD activation can be linearly compared across voxels of different activation levels and linearly related to the activities of individual neurons.

4. The results don't necessarily show decorrelation, relative to the unadapted representation, since the unadapted correlation between probe stimuli isn't measured. Instead the results show that within this stimulus sequence (random probes interleaved with repetitions of either A or B type stimuli) the correlations between similar stimuli (e.g. A1 and A2 probes) are lower if preceded by another similar stimulus (e.g. the A adaptor) than if preceded by a dissimilar stimulus (e.g. the B adaptor). This is interesting, but is not necessarily indicative of adaptation per se. For example, we know that representations of complex stimuli develop gradually over time, and that different category-related features become classifiable at different time points (e.g. reviewed in Contini, Wardle & Carlson (2017) <https://www.sciencedirect.com/science/article/pii/S0028393217300593>). In the present experiment, there is (presumably) a larger change in neural states required in the cases where the probe comes from a different vs same shape class as the adaptor. Could it be that when a larger distance must be travelled in "representational space" (e.g. for B probes after A adaptation) the nuances between nearby patterns (B1 vs B2 probes) do not have as much time to emerge, sharpen, and sustain themselves as when the distances travelled are very small (A to A1 or A to A2), and that this poorer differentiation manifests as higher correlation? That is, are we observing the effects of adaptation, or of a general hysteresis effect that would occur even in brains which did not adapt? I can't yet think of an experimental way of solving this problem so I don't consider it a flaw in this particular experimental design, but it would be interesting to hear the authors' thoughts on it.

Minor comments

- p6 line 108, "Four fMRI acquisitions were conducted using each of the shape classes as the adapting stimulus" --on first reading this sounded as if there were 4 adapting shape classes. Perhaps rephrase along the lines of "Four fMRI acquisitions were conducted using each of the two shape classes as the adapting stimulus, for a total of eight fMRI runs per participant."

- p14 line 290, misplaced comma in "(1-up, 3-down rule)"

- p15 line 317, should be "(larger or smaller)"

We would like to thank you for evaluating our manuscript “Adaptation decorrelates shape representations”, which we have revised in accordance with the reviewers’ suggestions and now resubmit for your consideration for publication in *Nature Communications*. In the pages that follow, we include our responses and the changes made point-by-point. For ease of reference, reviewer comments have been consecutively numbered and italicized.

To summarize the most substantive changes made to the manuscript:

- We present the various fMRI results separately for each stimulus set
- We report the significance map of the searchlight results assessed with a permutation test
- We have replaced the sharpening-vs-fatigue analysis with a new analysis and figure that relates response suppression to pattern decorrelation directly
- The Introduction and Discussion sections have been thoroughly revised to reflect reviewer suggestions

We note that all three reviewers expressed, directly or indirectly, an interest in comparing the neural and behavioral effects in the same individuals, so we include our response here. We also agree that it would be ideal if we could establish such a link. However, we believe that limits in the precision of any reasonable measure places this test out of reach.

To demonstrate our reasoning, we simulated data from N subjects exhibiting a perfect correlation between their decorrelation indices in behavior and fMRI. We then added Gaussian noise of the same magnitude as was observed in our data to each measurement. We found that, even if the two measures were perfectly correlated in reality (the ideal setting), measurement noise would require a sample size of over 200 subjects to have 80% chance of finding a $p < 0.05$ correlation. We should note that this limitation of between-measure correlations is one that afflicts many behavioral neuroscience studies, and is the subject of increased scrutiny in the “scientific replicability” community.

We thank you very much for the opportunity to incorporate these suggestions into our manuscript and resubmit our paper. Please do not hesitate to contact us if we can provide further clarification.

1 Reviewer 1

1.1 General remarks

The manuscript reports a study that uses fMRI to test the hypothesis that adaptation results in neural decorrelation and enhanced perceptual discriminability of similar stimuli. The study employs a highly controlled object space that allows the investigators to test this question in higher-level visual areas. The behavioral results are convincing. However, the fMRI findings are less conclusive.

In particular, the fMRI effect of decorrelation was only shown significantly in space B but not space A. Further methodological details are needed to assess how the voxel pattern similarity was estimated. Figure 3c- the key figure showing the effect of decorrelation is not convincing: most points are on the diagonal with error bars crossing the diagonal rather than shifted towards the y axis.

We thank the reviewer for this thoughtful critique of our work. Below we address each point raised individually.

We wish to note at the outset that it is indeed true that the measured effect is not significant for every individual subject, though this is rarely the case in any fMRI study. The error bars in the figure indicate the standard error of the mean across scan acquisitions within subject, suggesting some degree of intra-subject variability between measurements. Yet, our results show a consistent effect across all subjects, with a decreased voxel pattern similarity after adaptation to similar shapes. As a result, the statistics at the group level are highly significant ($t(9) = 5.0618$, $p = 0.0007$).

1.2 Comments

1.2.1 *The searchlight analysis shows patterns extending in the visual and parietal cortex; however, there is no mention of control analysis (e.g. permutation tests, comparison with shuffled labels) or significance testing (was there an arbitrary threshold chosen for the maps in Figure 3d?).*

Thanks for this suggestion. Our initial goal with the searchlight analysis was to show that the direction of the decorrelation effect was consistent and distributed through the cortex, which we believe is an important and unique result of our study and something that would have been difficult to obtain with other methodologies. We used an arbitrary threshold of $\Delta r = 0.10$ for this map.

The reviewer's suggestion is well taken, and we now also present a significance map showing the results of a permutation test shuffling the labels (using FSL randomise and Threshold-Free Cluster Enhancement). This figure shows that significant clusters are located on the occipital and ventral temporal lobes, with one additional cluster located on the superior temporal lobe. Notice that because the permutation test was performed in volumetric space, we modified the effect size map accordingly (i.e., combining data across subjects in volumetric space). The details of this analysis are included in the Methods section:

...We assessed significance at the whole-brain level using threshold-free cluster enhancement (TFCE; (Smith and Nichols, 2009)), an algorithm designed to offer the sensitivity of cluster-based thresholding without the need to set an arbitrary threshold. We corrected the TFCE map for familywise error rate using FSL's 1-sample group-mean permutation test (exhaustively testing all 1,024 permutations) and spatial 10 mm FWHM variance smoothing to reduce noise from poorly estimated variances in the permutation test procedure. Searchlight results are presented on the surface (Freesurfer's `fsaverage`) both with an effect size map (using a threshold of $\Delta r = 0.10$; Figure ??e) and with a significance map (using a threshold of $p = 0.01$; Figure ??f).

1.2.2 *Further, the behavioral and fMRI experiments were conducted on different participants. As a result, linking behavioral and fMRI findings is not possible. This compromises the link between perceptual discriminability and neural decorrelation due to adaptation. The behavioral effect was stronger for stimuli in space A rather than B. However, the fMRI study did not show a significant effect for space A. How can these findings be reconciled?*

We agree with the reviewer that conclusively linking the behavioral and fMRI findings would require that both data were collected from the same participants. However, as explained above, the sample size required for such a study would be prohibitively high. Given these constraints, our experiment instead focuses on relating the two effects by using a paradigm as similar as possible for both modalities. We then report that a similar effect is observed in both behavioral and fMRI experiments, suggesting that they are simultaneously present after adaptation.

The reviewer then suggests that “behavioral effect was stronger for stimuli in space A rather than B”, possibly alluding to the passage in the paper where we report that “we also observed a main effect of prototype stimulus on thresholds ($F(1, 77) = 14.4, p < 0.001$), suggesting that participants found it easier to discriminate stimuli within shape space “A” as compared to “B” (Figure 2b).” Notice however that this effect corresponds to baseline discriminability, and not the effect of decorrelation. In fact, we report in the following sentence that “mean decrease in threshold was 4.6 morph units for prototype A (one-sided t-test $t(19) = 4.8, p < 0.001$) and 6.4 units for prototype B ($t(19) = 8.8, p < 0.001$)”, i.e., the decorrelation effect was indeed stronger for prototype B, mirroring our fMRI results.

Therefore, the reviewer’s intuition is correct: There is indeed a larger fMRI decorrelation effect for the stimulus space that exhibits a larger behavioral effect. In the revised manuscript we point out this relationship:

... However, in post-hoc tests examining this effect in the two stimulus spaces separately, the correlation between the “B” probe stimuli (B1 and B2) was lower when subjects were adapted to shape class “B” than when they were adapted to shape class “A” ($t(9) = 4.83, p = 0.0009$), but the complementary test with the “A” shape class was not significant ($t(9) = 1.26, p = 0.24$). We note that the stronger decorrelation observed for shape class “B” is consistent with the larger decrease in perceptual discrimination thresholds for these stimuli reported above.

1.2.3 *An attentional control task was performed in the scanner. However, performance in this task was not reported. Therefore, it is difficult to assess whether attention was controlled for or whether differences in attention across conditions may drive stronger decorrelation of the fMRI signals. The searchlight analysis revealed parietal regions known to be involved in attentional processing. Was decorrelation in these regions due to attentional confounds? Stronger evidence is needed to discern whether neural decorrelation was due to adaptation rather than differential attention.*

We feel that it is difficult to conclude the existence of an attention effect simply from the presence of an fMRI difference in the parietal lobes (i.e., the reverse inference). Indeed, other studies have observed object-driven responses in dorsal regions (e.g. Grèzes and Decety, 2002; Denys et al., 2004). Our subjects performed a task for which the perceptual judgment was orthogonal to the key manipulation of stimulus similarity. Comparable tasks have been used in several prior experiments and subjects perform with uniformly high accuracy (Drucker and Aguirre, 2009; Murray et al., 2002; Fang et al., 2009); we did not record the performance of our subjects.

While we can’t exclude the possibility that “attention” (broadly considered) varies between the conditions of our experiment, the predicted effect upon our measurement would be the opposite of what we observed. The effects of attention on neural representations are found to be an *improved* signal-to-noise ratio — for instance, through a general upscaling of the activation without a systematic change in terms of narrowing or

decrease in variability (McAdams and Maunsell, 1999), or through a narrower position tuning at the neural population level (Fischer and Whitney, 2009). It is difficult to imagine, therefore, how the “novel”, different stimuli would evoke patterns that have a relatively greater correlation (i.e., less decorrelation).

1.2.4 *Two sets of vertices are separated based on univariate response. Both these clusters are shown to relate to decorrelation. However, no statistical comparison is reported that tests which of the two clusters shows stronger decorrelation. This is important for assessing whether the lower signal to noise ratio accounts for reduced correlation. Further, the two clusters may have different variance, but this is not taken into account when comparing these voxels; yet difference in the variance may affect the decorrelation analysis.*

Thanks for this great suggestion. When comparing both clusters, we find only marginal support for the proposal that decorrelation is stronger for the 50 vertices undergoing most attenuation ($t(9) = 2.2049$, $p = 0.0549$). We also compared the two clusters in terms of decorrelation standard deviation across runs, but observed no significant differences ($t(9) = 0.9186$, $p = 0.3823$).

The reviewer’s comment prompted us to undertake a further analysis in which we systematically related the degree of decorrelation to the size and direction of response modulation. See below for a related paragraph from the manuscript.

We find, however, that this reduction in amplitude varies markedly across voxels. We calculated the magnitude of response suppression for each voxel as a scaling factor between adapted and unadapted responses, and then identified sets of voxels with different degrees of response suppression. The 50 voxels with lowest values for this index had a mean suppression value of 0.82 ± 0.05 (i.e., the BOLD fMRI signal evoked by the stimuli was reduced on average by 18% in the adapted condition). In contrast, the 50 voxels with the largest values for this index actually demonstrated response *enhancement* in the adaptation condition (1.17 ± 0.08 , or an increase by 17% of response amplitude in the adaptation condition). If a reduction in response amplitude alone accounts for the decorrelation of patterns that we observe in the adaptation condition, then a decorrelation effect should not be present in the subset of voxels with response enhancement. In disagreement with this account, a significant decorrelation effect was still found (two-sample t -test on Fisher-Z transformed correlation coefficients: $t(9) = 2.50$, $p = 0.034$), although decorrelation was marginally stronger for the subset of voxels with most suppression ($t(9) = 2.2049$, $p = 0.055$). To analyze more closely the relationship between the suppressive effect of adaptation and the decorrelation effect, we grouped voxels into various bins according to the degree of response suppression (scaling factor) exhibited within a scan. We then computed the degree of pattern decorrelation for each bin containing 10 or more voxels. We found that decorrelation effect is largest in voxels whose responses are most suppressed, suggesting that this effect is directly or indirectly linked to the ubiquitous reduction in the amplitude of evoked responses (Figure ??g), although response suppression alone cannot account for the observed pattern decorrelation effects.

1.2.5 *An attempt is made to compare the results with other models of adaptation: sharpening vs. fatigue. Again here two sets of vertices are compared; however the statistics are not convincing (why is 1-tail t-test used?). Further, previous imaging papers have used multivariate analysis and compared voxel-tuning curves to provide evidence for sharpening. Further analysis of the data following this methodology should be conducted to conclusively infer whether decorrelation relates to sharpening rather than fatigue. The data currently presented remain unconvincing in discerning between the neural mechanisms that underlie adaptation.*

We appreciate the reviewer’s concerns, which have been echoed also by Reviewer 3 in Comment 3.2.3. Given this feedback, we concede that our analyses were unable to adjudicate between neural sharpening and

fatigue. Indeed, the inability of fMRI to distinguish between these two models at the neural level have been addressed recently (Alink et al., 2017).

In light of these observations, we instead propose to examine the mechanism underlying decorrelation by examining its relationship to repetition suppression. In the revised manuscript, we examine more directly how these two effects of adaptation relate to one another, and find that subsets of voxels undergoing larger response suppression also exhibit greater pattern decorrelation.

2 Reviewer 2

2.1 General remarks

Mattar and colleagues present a careful and well-conducted study investigating the impact of prior exposure on both behavioral shape judgments and BOLD fMRI responses in shape-selective cortical regions. This issue has been much debated in recent years and major strengths of this study are the combination of behavioral and fMRI experiments and the elegant and simple design that precisely targets the potential role of decorrelated neural responses.

However, I think the manuscript can be strengthened by expanding the discussion of relevant literature and extending the presentation of some of the results. The sample size for the fMRI experiment is relatively small ($n = 10$), but the authors show the results at an individual subject level (e.g. Figure 3C) and I don't think this is a major concern.

Thank you.

2.2 Comments

2.2.1 *The authors use the terminology of ‘adaptation’, but the impact of prior exposure on neural responses has also been discussed in terms of ‘repetition suppression’ (particularly in the neuroimaging field) and even ‘habituation’. There is much literature trying to relate repetition suppression of the BOLD signal to behavior and while the authors paradigm assumes a short time frame for ‘adaptation’, repetition suppression effects have been shown to last longer than 24 hours (e.g. work by Alex Martin and colleagues). I think the manuscript would benefit from a broader discussion of this prior literature, placing the current results more in context and broadening the current focus (particularly in the introduction) from primarily neurophysiological studies (especially given that the current study uses fMRI).*

Thanks for this feedback. In the revised document we explore the relationship between our study and prior work on repetition suppression:

Neural responses to visual stimuli are modulated by the preceding temporal context, a phenomenon known as “adaptation” (Enroth-Cugell and Shapley, 1973; Ohzawa et al., 1982; Dragoi et al., 2000; Kohn and Movshon, 2004; Engel, 2005; Krekelberg et al., 2006; Kusunoki et al., 2006; Schwartz et al., 2007; Wark et al., 2009). Adaptation is often manifested as a reduction in the neural response evoked by stimuli that are identical or similar to those observed previously. This effect is observed in various brain regions and over a wide range of timescales — from milliseconds (Sobotka and Ringo, 1996) to minutes (Henson et al., 2000) to days (van Turennout et al., 2000) — suggesting that this process is constantly at work in the nervous system (Mattar et al., 2016).

We have also expanded the Discussion accordingly:

Our paradigm bears some resemblance to category learning, especially as discussed in prototype theory (e.g. Ashby and Maddox, 2005), though it differs in two important ways. First, instead of learning to discriminate A from B, our subjects had to discriminate *within* category A or B, which is a question not typically addressed in category learning studies (although see de Beeck et al. 2006). Second, our study examines how discrimination is affected by the preceding few seconds and minutes of exposure, a timescale shorter than often considered in category learning studies. Our study also shares similarities with perceptual learning paradigms, in which subjects

learn to discriminate between two initially indistinguishable stimuli after several hours or days of practice (Goldstone and Gibson, 1962). While the phenomena of category and perceptual learning manifest over much longer time scales than the seconds-to-minutes of adaptation studied here, we note that adaptation and long timescale perceptual representations may share underlying neural mechanisms (Mattar et al., 2016), in line with observations of repetition suppression over intervals as long as multiple days (van Turennout et al., 2000).

2.2.2 *The authors found a difference in thresholds between the two stimulus spaces (easier discrimination for A), a difference in the decrease of behavioral thresholds (greater decrease for B), and no difference in correlation of BOLD responses for stimulus space A when adapted/unadapted. However, I could find no discussion of how these results might relate to each other — I think such a discussion would be worth adding.*

Thanks for this suggestion. We now address this point in the Discussion section, as seen below:

... However, in post-hoc tests examining this effect in the two stimulus spaces separately, the correlation between the “B” probe stimuli (B1 and B2) was lower when subjects were adapted to shape class “B” than when they were adapted to shape class “A” ($t(9) = 4.83, p = 0.0009$), but the complementary test with the “A” shape class was not significant ($t(9) = 1.26, p = 0.24$). We note that the stronger decorrelation observed for shape class “B” is consistent with the larger decrease in perceptual discrimination thresholds for these stimuli reported above.

2.2.3 *Given the different results for sets A and B in the fMRI data, I think the authors should show the consistency of the effects across subjects for each set separately.*

A great suggestion. In the revised manuscript we include the results across subjects for each set separately. The reported effect in this ROI is significant for Probe B ($t(9) = 4.83, p = 0.0009$) but not for Probe A ($t(9) = 1.26, p = 0.24$). This difference can also be seen in the new panel added to Figure 3d, showing the effects across subjects for each set separately.

We also examined the consistency of the decorrelation effect across subjects. For each subject, we measured the Pearson correlation of the voxel responses evoked by a pair of probe stimuli when those stimuli were from the same class as the adaptors, and when they were from the unadapted shape class (Figure ??c). All ten subjects had a lower correlation between probe stimuli in the adapted condition (paired t-test on Fisher-Z transformed correlation coefficients: $t(9) = 5.06, p < 0.0007$). Examining the effect in the two stimulus spaces separately we found that 6 out of 10 subjects had a lower correlation between “A” probe stimuli in the adapted condition, and that 9 out of 10 subjects had a lower correlation between “B” probe stimuli in the adapted condition (Figure ??d).

2.2.4 *Similarly, I think the authors should report the BOLD amplitudes separately for sets A and B.*

Another good suggestion. We now report the BOLD amplitudes separately for both sets:

As adaptation is known to reduce the amplitude of BOLD response, one possible mechanism for our findings is a reduction in stimulus-evoked responses. If this reduction in response occurs in the setting of independent, unchanged measurement noise, the reduced correlation we observe in the patterns evoked by adapted stimuli may be the product only of a lower signal-to-noise ratio. Consistent with this mechanism, we find that the evoked BOLD fMRI signal amplitude is smaller in the adapted as compared to the unadapted condition within left LO (percent signal change in adapted condition: $1.30\% \pm 0.16$ vs. unadapted: $1.45\% \pm 0.16$; paired t -test: $t(9) = 2.78, p = 0.0213$). A similar response reduction was observed for either shape class (percent signal change for shape class “A“ in adapted condition: $1.28\% \pm 0.15$ vs. unadapted: $1.38\% \pm 0.15$; percent signal change for shape class “B“ in adapted condition: $1.33\% \pm 0.19$ vs. unadapted: $1.51\% \pm 0.21$).

2.2.5 *The analysis in which the authors selected the vertices showing a response enhancement is very nice, but it would be helpful to compute the decorrelation effect for the vertices showing the strongest response reduction as well for comparison (i.e. is there any difference in the decorrelation effect between these sets of vertices).*

Thanks for this suggestion, which has also been made by Reviewer 1 in Comment 1.2.4. As seen above, when comparing both clusters, we see decorrelation is stronger for the 50 vertices undergoing most attenuation, though the effect is only marginally significant ($t(9) = 2.2049, p = 0.0549$). We also compared the two clusters in terms of decorrelation standard deviation across runs, but observed no significant differences ($t(9) = 0.9186, p = 0.3823$).

Below we reproduce the relevant portion of the Results section reporting this comparison.

We find, however, that this reduction in amplitude varies markedly across voxels. We calculated the magnitude of response suppression for each voxel as a scaling factor between adapted and unadapted responses, and then identified sets of voxels with different degrees of response suppression. The 50 voxels with lowest values for this index had a mean suppression value of 0.82 ± 0.05 (i.e., the BOLD fMRI signal evoked by the stimuli was reduced on average by 18% in the adapted condition). In contrast, the 50 voxels with the largest values for this index actually demonstrated response *enhancement* in the adaptation condition (1.17 ± 0.08 , or an increase by 17% of response amplitude in the adaptation condition). If a reduction in response amplitude alone accounts for the decorrelation of patterns that we observe in the adaptation condition, then a decorrelation effect should not be present in the subset of voxels with response enhancement. In disagreement with this account, a significant decorrelation effect was still found (two-sample t -test on Fisher-Z transformed correlation coefficients: $t(9) = 2.50, p = 0.034$), although decorrelation was marginally stronger for the subset of voxels with most suppression ($t(9) = 2.2049, p = 0.055$).

It would be interesting to know how the correlation between sets A and B (and not just within set) is affected by adaptation.

We appreciate this suggestion, but we note that examining the correlation between sets is difficult given that one of the sets is always “adapted”. To get around this issue, we performed the same analyses by pulling voxel patterns across runs, which allowed us to examine correlations between sets in both adapted and unadapted states.

Specifically, we averaged the evoked voxel responses for each adapted and unadapted stimulus across scan runs. To examine the validity of this approach, we first computed the correlation between the average responses across runs for the two probe stimuli. We found that the correlation between evoked responses

after adapting to the same space was apparently lower than after adapting to the other space, though this effect was not significant (Figure R1a).

We then computed the correlation between sets A and B. We found a trend where the correlation between probes from set “A” and probes from set “B” was apparently higher when only one of the two shapes was adapted than when both or neither were adapted (compare middle bars vs. left/right bars in Figure R1b), though this difference was not significant (ANOVA: $F(1, 9) = 1.6994, p = 0.2247$). Notice that if Probe “A” is adapted and Probe “B” is unadapted, both are adapted to shape set “A”. Similarly, if Probe “A” is unadapted and Probe “B” is adapted, both are adapted to shape set “B”. The existence of a common signal from the adapting phase is likely responsible for the apparently higher between-run correlation when only one shape set is adapted (middle bars in Figure R1b). Importantly, notice that this “leaking” of an adaptation signal onto the probe phase cannot explain the results in the main paper, given that it would produce an effect opposite to the one we reported.

Figure R1. Voxel pattern correlation between runs. *a)* Correlation between the average evoked pattern across runs for the two probes *within* a set. *b)* Correlation between the average evoked pattern across runs for the two probes *between* sets. *Left:* Correlation between probes A and B when both are adapted; *Middle:* Correlation between probes A and B when either A or B are adapted; *Right:* Correlation between probes A and B when neither is adapted.

2.2.6 *Given that the authors are collecting BOLD fMRI data, I think they should be more cautious in their use of ‘neural’. For example, at the start of the discussion, I don’t think it is appropriate to say, “We used neural data”. The same concern applies elsewhere in the manuscript.*

Thanks for this suggestion. Indeed, the word “neural” is potentially misleading, in particular to distinguish it from previous studies where decorrelation is discussed in terms of neuronal firing rates. We have edited the manuscript throughout to use “neuroimaging data”, “voxel pattern”, and “BOLD amplitude”, in places where we used the term “neural” in reference to fMRI data.

2.2.7 *The authors point out that trial-by-trial data would be nice, but the current data is limited not just because such trial-by-trial analyses are not possible, but more simply because the behavioral and BOLD data were not collected simultaneously and were even collected in different subjects.*

Indeed. We have edited the corresponding passage of the Discussion:

While our study demonstrates perceptual improvements in discrimination performance and voxel changes in representation using the same adaptation procedure, we note that our findings do not directly relate these phenomena. An ideal model would provide a quantitative mapping

between neural and perceptual effects on a trial-by-trial basis. A challenge to such an effort is that measurement of the voxel responses is complicated in the face of a perceptual task that requires the subject to explicitly process the similarity of presented stimuli, since the behavioral task could produce confounding effects in the neural data. Alternatively, if decorrelation is a stable property of the individual and with sufficient inter-subject variability, a link could be established by measuring both perceptual and neural effects on the same individuals and examining whether the effects co-vary. A complete model would also account for the cortical extent over which this decorrelation effect is observed. We find that visual cortex broadly demonstrates the decorrelation effect, although it is of greater strength in object-responsive areas that have been previously shown to exhibit coarse spatial coding for object shape (Drucker and Aguirre, 2009).

2.2.8 *The authors briefly mention category learning studies and suggest that in such studies the discrimination is typically between- and not within-category as in the current study. However, I think the study by Op de Beeck and colleagues (2006, J. Neurosci) involved within-category learning with an analytical focus on the correlation of responses to the different categories.*

Thanks for pointing out this very relevant study which we had not cited in our initial submission. The revised manuscript now mentions this study in the Discussion section:

Our paradigm bears some resemblance to category learning, especially as discussed in prototype theory (e.g. Ashby and Maddox, 2005), though it differs in two important ways. First, instead of learning to discriminate A from B, our subjects had to discriminate *within* category A or B, which is a question not typically addressed in category learning studies (although see de Beeck et al. 2006). Second, our study examines how discrimination is affected by the preceding few seconds and minutes of exposure, a timescale shorter than often considered in category learning studies. Our study also shares similarities with perceptual learning paradigms, in which subjects learn to discriminate between two initially indistinguishable stimuli after several hours or days of practice (Goldstone and Gibson, 1962). While the phenomena of category and perceptual learning manifest over much longer time scales than the seconds-to-minutes of adaptation studied here, we note that adaptation and long timescale perceptual representations may share underlying neural mechanisms (Mattar et al., 2016), in line with observations of repetition suppression over intervals as long as multiple days (van Turennout et al., 2000).

3 Reviewer 3

3.1 General remarks

The paper presents a very nice hypothesis-driven use of MVPA with a carefully-constructed stimulus space, to test the computational hypothesis that one of the effects of adaptation is to decorrelate neural responses to stimuli similar to the adaptor. The paper was clear, succinct, and enjoyable to read.

Two experiments are performed, in different groups of observers, one psychophysical and one using fMRI. Both use shape stimuli drawn from a 24-dimensional generative shape space. Adapting stimuli are drawn from the regions closely surrounding one of two arbitrary "prototype" points in the space. In the psychophysical experiment, perceptual shape aftereffects are quantified by measuring discrimination thresholds in a delayed 2AFC match to sample task along a 1D trajectory in the shape space connecting the two prototype shapes, after adapting to one or the other prototype regions. A similar protocol is used in the fMRI experiment, in which participants adapt to one of the two prototype regions (blocked across runs) and are shown one of four probe stimuli, two each drawn from the areas near the two prototypes (probe stimuli randomly interleaved within each run). The Pearson correlation between BOLD patterns evoked by pairs of stimuli drawn from the adapted vs unadapted regions in shape space are then calculated and compared. The main result of the paper is that shapes from an adapted region of shape space evoke less correlated patterns than do shapes from an unadapted region, which is consistent with the possibility that responses of individual neurons become less correlated following adaptation.

We thank the reviewer for the very generous comments about our writing and for the detailed summary of our work. Below, we address each comment individually.

3.2 Comments

3.2.1 *The evidence for a causal link between the observed neural pattern decorrelation and behavioural adaption aftereffect could have been substantially strengthened by showing a positive relationship between the degree of pattern decorrelation and perceptual aftereffect in the same individuals. The variability in both of these effects was reasonably large across individuals, but unfortunately the same participants were not used in the fMRI and psychophysical experiments so this test cannot be done.*

Indeed, this is a major feature missing from this paper, and we certainly share the reviewer's interest in such data. However, as explained in the beginning of this letter, the sample size required for such a study would be prohibitively high. Given these constraints, our experiment instead focuses on relating the two effects by using a paradigm as similar as possible for both modalities. We then report that a similar effect is observed in both behavioral and fMRI experiments, suggesting that they are simultaneously present after adaptation.

3.2.2 *There is some similar previous work which is not discussed. In particular, I am aware of Castaldi et al. (2016) "Effects of adaptation on numerosity decoding in the human brain" NeuroImage (<https://www.ncbi.nlm.nih.gov/>) That study presents dot displays of various numerosities and uses a linear support vector machine to classify numerosity from fMRI-measured cortical patterns. They find that reasonable classification is achievable both before and after adaptation, but that adaptation improves decodability (consistent with the report here of decreased correlation between patterns after adaptation). They also test the brain-behaviour link described in the point above, showing that improvement in decodability after adaptation is correlated across participants with the behavioural numerosity aftereffect. Since that study examines a different modality and brain region of interest, and uses the different-but-related metric of SVM-classifiability rather than correlation, it does not undermine the novelty of the results presented here. However, it does reduce the apparent methodological innovation, and deserves mention in the Introduction.*

Thanks for this suggestion. We were not aware of this paper but now refer to it in both the Introduction and Discussion, as reproduced below.

Adaptation has been proposed to facilitate efficient sensory coding by tuning the response properties of neural populations to the current sensory environment (Barlow and Földiák, 1989; Clifford et al., 2007; Kohn, 2007). In particular, adaptation may reduce the correlation between neural activity patterns corresponding to frequently encountered stimuli (Barlow and Földiák, 1989; Barlow, 1990), either by shifting neuronal tuning curves away from one another, or by narrowing neuronal selectivity (Clifford et al., 2000; Kohn, 2007; Grill-Spector et al., 2006; Seriès et al., 2009; Cortes et al., 2012). Empirical support for this hypothesis has been found in some animal studies: neurophysiological recordings from monkey primary visual cortex show that adaptation to stimulus orientation decorrelates neural responses (Mueller et al., 1999; Gutnisky and Dragoi, 2008), and recordings from cat primary visual cortex show that adaptation promotes population homeostasis (Benucci et al., 2013). In humans, adaptation improves fMRI decoding of numerosity in the intraparietal sulcus (Castaldi et al., 2016).

Also in the Discussion:

...Our results offer three novel contributions: (i) we provide evidence for enhanced perceptual discriminability of high-level stimuli (3D shapes) following adaptation, thus clarifying earlier findings whose results were mixed (Rhodes et al., 2007; Ng et al., 2008; Oruç et al., 2011; Keefe et al., 2013) and building on previous findings from a different domain (Castaldi et al., 2016); (ii) we demonstrate a decorrelation of voxel pattern representations in human observers undergoing visual adaptation; (iii) we offer joint behavioral and neuroimaging evidence using a similar experimental paradigm, thus offering a link between the neural effect and its behavioral consequences.

3.2.3 *The sharpening vs fatigue analysis on page 10 does not seem to be a good test of those hypotheses. The prediction of the sharpening hypothesis is that neurons that respond more weakly to begin with to the adapting stimuli are most suppressed by adaptation. But here, we don't have measures of the patterns elicited by the adapting stimuli before and after adaptation. Instead (if I understand correctly) vertices are categorised according to their response after adaptation, to either adapted or non-adapted probe shapes. To consider a simpler analogy: imagine we measure V1 single-cell responses to vertical and 45 degree gratings after adapting to vertical gratings, and find that firing rates are on average lower to vertical test gratings than 45 degree ones. We now categorise the cells based on their response (after adapting to vertical) to the unadapted 45 degree gratings. We find that the cells that respond most weakly to 45 degree gratings after vertical adaptation are those most dramatically low in activity when shown vertical gratings. In this case, it seems clear that this result cannot be interpreted either for or against the sharpening hypothesis; the neurons responding weakly to the unadapted stimulus may also have had a low response to the adapting stimulus prior to adaptation (e.g. because they prefer horizontal orientations) in which case their suppressed vertical response would be consistent with sharpening, or they may have had high responses to the adaptor prior to adaptation (e.g. they are tuned to vertical) in which case the results are simply demonstrating adaptation. In other words, because baseline responsiveness is unknown, the analysis seems to confound stimulus preference with the effect of adaptation in difficult to interpret ways. It only makes sense if we assume that those neurons which respond weakly to the non-adapted stimulus also responded weakly to the adapted stimulus prior to adaptation. Finally, it also seems questionable to assume that differences in estimated BOLD activation can be linearly compared across voxels of different activation levels and linearly related to the activities of individual neurons.*

We appreciate the reviewer's concerns, also shared by Reviewer 1 in Comment 1.2.5. Given the feedback from both, we concede that our analyses was unable to adjudicate between neural sharpening and fatigue.

Indeed, the inability of fMRI to distinguish between these two models at the neural level have been addressed recently (Alink et al., 2017).

In light of these observations, we instead propose to examine the mechanisms underlying decorrelation by examining its relationship to repetition suppression. In the revised manuscript, we examine more directly how these two effects of adaptation relate to one another, and find that subsets of voxels undergoing larger response suppression also exhibit greater pattern decorrelation. The relevant passage in the Results section is reproduced below:

... To analyze more closely the relationship between the suppressive effect of adaptation and the decorrelation effect, we grouped voxels into various bins according to the degree of response suppression (scaling factor) exhibited within a scan. We then computed the degree of pattern decorrelation for each bin containing 10 or more voxels. We found that decorrelation effect is largest in voxels whose responses are most suppressed, suggesting that this effect is directly or indirectly linked to the ubiquitous reduction in the amplitude of evoked responses (Figure ??g), although response suppression alone cannot account for the observed pattern decorrelation effects.

3.2.4 *The results don't necessarily show decorrelation, relative to the unadapted representation, since the unadapted correlation between probe stimuli isn't measured. Instead the results show that within this stimulus sequence (random probes interleaved with repetitions of either A or B type stimuli) the correlations between similar stimuli (e.g. A1 and A2 probes) are lower if preceded by another similar stimulus (e.g. the A adaptor) than if preceded by a dissimilar stimulus (e.g. the B adaptor). This is interesting, but is not necessarily indicative of adaptation per se. For example, we know that representations of complex stimuli develop gradually over time, and that different category-related features become classifiable at different time points (e.g. reviewed in Contini, Wardle & Carlson (2017) <https://www.sciencedirect.com/science/article/pii/S0028393217300593>). In the present experiment, there is (presumably) a larger change in neural states required in the cases where the probe comes from a different vs same shape class as the adaptor. Could it be that when a larger distance must be travelled in "representational space" (e.g. for B probes after A adaptation) the nuances between nearby patterns (B1 vs B2 probes) do not have as much time to emerge, sharpen, and sustain themselves as when the distances travelled are very small (A to A1 or A to A2), and that this poorer differentiation manifests as higher correlation? That is, are we observing the effects of adaptation, or of a general hysteresis effect that would occur even in brains which did not adapt? I can't yet think of an experimental way of solving this problem so I don't consider it a flaw in this particular experimental design, but it would be interesting to hear the authors' thoughts on it.*

This is a great point, and we completely agree with the reviewer's intuition. At the heart of this issue is what we and the reviewer mean by "adaptation". We tend to think about adaptation broadly as a set of mechanisms operating at different levels of the neural hierarchy and at different timescales with the aim to optimize sensory processing in face of environmental changes (Mattar et al., 2016). In particular, integrating sensory information over the adaptation phase in our study can be thought of as a shift in "representational space", such that responses evoked by adapted probes are both weaker (due to a smaller distance travelled) and more differentiated (due to a greater difference in vector direction). Equivalently, responses evoked by the non-adapted probes are more confusable due to being distant from the integrated sensory context, with this confusability manifesting as higher correlation (notice that Pearson correlation can be understood as the angle between two mean-centered vectors). So, in this view, we are observing the effects of adaptation — integrated sensory context — both in the univariate magnitude of the evoked responses and in the multivariate evoked patterns.

3.2.5 *p6 line 108, “Four fMRI acquisitions were conducted using each of the shape classes as the adapting stimulus” –on first reading this sounded as if there were 4 adapting shape classes. Perhaps rephrase along the lines of “Four fMRI acquisitions were conducted using each of the two shape classes as the adapting stimulus, for a total of eight fMRI runs per participant.”*

A great suggestion, thanks. We now use the suggested phrasing in the revised document.

3.2.6 *p14 line 290, misplaced comma in “(1-up, 3-down rule)”*

Fixed.

3.2.7 *p15 line 317, should be “(larger or smaller)”*

Fixed.

References

- Alink, A., Abdulrahman, H., and Henson, R. N. (2017). From neurons to voxels-repetition suppression is best modelled by local neural scaling. *bioRxiv*, page 170498.
- Ashby, F. G. and Maddox, W. T. (2005). Human category learning. *Annual review of psychology*, 56(1):149–78.
- Barlow, H. B. (1990). A theory about the functional role and synaptic mechanism of visual after-effects. In Blakemore, C. B., editor, *Vision: coding and efficiency*, pages 363–375. Cambridge University Press, Cambridge.
- Barlow, H. B. and Földiák, P. (1989). Adaptation and decorrelation in the cortex. In Durbin, R., Miall, C., and Mitchison, G., editors, *The Computing Neuron*, number 1988, pages 54–72. T. J. Press, Cornwall.
- Benucci, A., Saleem, A. B., and Carandini, M. (2013). Adaptation maintains population homeostasis in primary visual cortex. *Nat Neurosci*, 16(6):724–729.
- Castaldi, E., Aagten-Murphy, D., Tosetti, M., Burr, D., and Morrone, M. C. (2016). Effects of adaptation on numerosity decoding in the human brain. *NeuroImage*, 143:364–377.
- Clifford, C. W. G., Webster, M. A., Stanley, G. B., Stocker, A. A., Kohn, A., Sharpee, T. O., and Schwartz, O. (2007). Visual adaptation: Neural, psychological and computational aspects. *Vision Research*, 47(25):3125–3131.
- Clifford, C. W. G., Wenderoth, P., and Spehar, B. (2000). A functional angle on some after-effects in cortical vision. *Proceedings. Biological sciences / The Royal Society*, 267(1454):1705–1710.
- Cortes, J. M., Marinazzo, D., Series, P., Oram, M. W., Sejnowski, T. J., and Van Rossum, M. C. (2012). The effect of neural adaptation on population coding accuracy. *Journal of computational neuroscience*, 32(3):387–402.
- de Beeck, H. P. O., Baker, C. I., DiCarlo, J. J., and Kanwisher, N. G. (2006). Discrimination training alters object representations in human extrastriate cortex. *Journal of Neuroscience*, 26(50):13025–13036.

- Denys, K., Vanduffel, W., Fize, D., Nelissen, K., Peuskens, H., Essen, D. V., and Orban, G. A. (2004). The processing of visual shape in the cerebral cortex of human and nonhuman primates: a functional magnetic resonance imaging study. *Journal of Neuroscience*, 24(10):2551–2565.
- Dragoi, V., Sharma, J., and Sur, M. (2000). Adaptation-induced plasticity of orientation tuning in adult visual cortex. *Neuron*, 28(1):287–298.
- Drucker, D. M. and Aguirre, G. K. (2009). Different spatial scales of shape similarity representation in lateral and ventral LOC. *Cerebral Cortex*, 19(10):2269–2280.
- Engel, S. A. (2005). Adaptation of oriented and unoriented color-selective neurons in human visual areas. *Neuron*, 45(4):613–623.
- Enroth-Cugell, C. and Shapley, R. (1973). Adaptation and dynamics of cat retinal ganglion cells. *Journal of Physiology*, 233:271–309.
- Fang, F., Boyaci, H., and Kersten, D. (2009). Border ownership selectivity in human early visual cortex and its modulation by attention. *The Journal of neuroscience : the official journal of the Society for Neuroscience*, 29(2):460–5.
- Fischer, J. and Whitney, D. (2009). Attention narrows position tuning of population responses in v1. *Current biology*, 19(16):1356–1361.
- Goldstone, R. L. and Gibson, E. J. (1962). Perceptual Learning. *Learning*, 49(1):585–612.
- Grèzes, J. and Decety, J. (2002). Does visual perception of object afford action? Evidence from a neuroimaging study. *Neuropsychologia*, 40(2):212–222.
- Grill-Spector, K., Henson, R., and Martin, A. (2006). Repetition and the brain: neural models of stimulus-specific effects. *Trends in cognitive sciences*, 10(1):14–23.
- Gutnisky, D. A. and Dragoi, V. (2008). Adaptive coding of visual information in neural populations. *Nature*, 452(7184):220–4.
- Henson, R., Shallice, T., and Dolan, R. (2000). Neuroimaging evidence for dissociable forms of repetition priming. *Science*, 287(5456):1269–1272.
- Keefe, B. D., Dzhelyova, M., Perrett, D. I., and Barraclough, N. E. (2013). Adaptation improves face trustworthiness discrimination. *Frontiers in psychology*, 4(June):358.
- Kohn, A. (2007). Visual Adaptation: Physiology, Mechanisms, and Functional Benefits. *Journal of Neurophysiology*, 10461:3155–3164.
- Kohn, A. and Movshon, J. A. (2004). Adaptation changes the direction tuning of macaque MT neurons. *Nature neuroscience*, 7(7):764–72.
- Krekelberg, B., van Wezel, R. J. A., and Albright, T. D. (2006). Adaptation in macaque MT reduces perceived speed and improves speed discrimination. *Journal of neurophysiology*, 95(1):255–70.
- Kusunoki, M., Moutoussis, K., and Zeki, S. (2006). Effect of background colors on the tuning of color-selective cells in monkey area V4. *Journal of Neurophysiology*, 95:3047–3059.
- Mattar, M. G., Kahn, D. A., Thompson-Schill, S. L., and Aguirre, G. K. (2016). Varying timescales of stimulus integration unite neural adaptation and prototype formation. *Current Biology*, 26(13):1669–1676.
- McAdams, C. J. and Maunsell, J. H. (1999). Effects of attention on orientation-tuning functions of single neurons in macaque cortical area v4. *Journal of Neuroscience*, 19(1):431–441.

- Mueller, J. R., Metha, A. B., Krauskopf, J., and Lennie, P. (1999). Rapid Adaptation in Visual Cortex to the Structure of Images. *Science*, 285(5432):1405–1408.
- Murray, S. O., Kersten, D., Olshausen, B. a., Schrater, P., and Woods, D. L. (2002). Shape perception reduces activity in human primary visual cortex. *Proceedings of the National Academy of Sciences of the United States of America*, 99(23):15164–9.
- Ng, M., Boynton, G. M., and Fine, I. (2008). Face adaptation does not improve performance on search or discrimination tasks. *Journal of Vision*, 8(1):1.1–20.
- Ohzawa, I., Sclar, G., and Freeman, R. D. (1982). Contrast gain control in the cat visual cortex. *Nature*, 298(5871):266–268.
- Oruç, I., Barton, J. J. S., Oruc, I., and Barton, J. J. S. (2011). Adaptation improves discrimination of face identity. *Proceedings. Biological sciences / The Royal Society*, 278(1718):2591–2597.
- Rhodes, G., Maloney, L. T., Turner, J., and Ewing, L. (2007). Adaptive face coding and discrimination around the average face. *Vision Research*, 47(7):974–989.
- Schwartz, O., Hsu, A., and Dayan, P. (2007). Space and time in visual context. *Nature reviews. Neuroscience*, 8(7):522–535.
- Seriès, P., Stocker, A. A., and Simoncelli, E. P. (2009). Is the homunculus ”aware” of sensory adaptation? *Neural Computation*, 21(12):3271–3304.
- Smith, S. M. and Nichols, T. E. (2009). Threshold-free cluster enhancement: addressing problems of smoothing, threshold dependence and localisation in cluster inference. *Neuroimage*, 44(1):83–98.
- Sobotka, S. and Ringo, J. L. (1996). Mnemonic responses of single units recorded from monkey inferotemporal cortex, accessed via transcommissural versus direct pathways: a dissociation between unit activity and behavior. *Journal of Neuroscience*, 16(13):4222–4230.
- van Turennout, M., Ellmore, T., and Martin, A. (2000). Long-lasting cortical plasticity in the object naming system. *Nature neuroscience*, 3(12):1329.
- Wark, B., Fairhall, A., and Rieke, F. (2009). Timescales of Inference in Visual Adaptation. *Neuron*, 61(5):750–761.

REVIEWERS' COMMENTS:

Reviewer #1 (Remarks to the Author):

The manuscript addresses an interesting question. However, the results presented appear weak and the evidence for a link between decorrelation and adaptation remains inconclusive.

1. The decorrelation effect appears to be weak. This effect was tested on a relatively small sample of participants (n=10), and only a subset of these participants showed decorrelation. I appreciate that fMRI data from individual participants may not be significant; however Figure 3c demonstrates a rather weak effect of decorrelation as most data points are on the diagonal rather than shifted towards the y axis (only 3-4 data points appear to be shifted from the diagonal). Further the authors do not explain why the fMRI effect of decorrelation was significant only for stimulus space B but not space A. Replication of the effect across stimulus spaces would strengthen the evidence for this rather weak effect.

2. The evidence provided by the authors regarding the link between fMRI and behavioral data is rather indirect. The authors write: 'There is indeed a larger fMRI decorrelation effect for the stimulus space that exhibits a larger behavioral effect.' However, the lack of both behavioral and fMRI data on the same individuals undermines the evidence for this link. I appreciate the constraints related to data collection; however, conclusions about neural mechanisms (i.e. decorrelation) that underlie behavior cannot be made without data-driven evidence.

3. The authors question the possible role of attention in decorrelation. However, several neurophysiology studies (e.g. Reynolds) provide evidence that attention decorrelates neural representations in the ventral visual stream (e.g. area V4). Again here, the lack of data undermines the conclusions of the study. The authors state that they did not collect participant responses for the attentional task. Therefore, it remains unclear whether the decorrelation observed relates to differences in attention or adaptation across stimulus conditions.

4. Finally, the manuscript fails to provide a mechanistic account and differentiate between mechanisms of decorrelation; that is, sharpening vs. fatigue. Instead the authors relate response suppression to decorrelation. This is interesting but it does not advance our understanding of the neural mechanisms that underlie adaptation.

Reviewer #2 (Remarks to the Author):

The authors have fully addressed all my concerns and I thank them for providing the additional data. I have no further comments. A very nice study.

Reviewer #3 (Remarks to the Author):

Thank you to the authors for an extremely comprehensive and considered set of responses to the three reviewers. The revised manuscript is stronger, and the new repetition suppression analysis is interesting.

As Reviewer 1 points out, since the decorrelation effect is small it would be great to replicate it in a larger set of observers and/or with different shape stimuli. I'm not sure this is a fatal flaw though, since the standard of evidence meets that of much other published literature.

I remain not fully convinced that the decorrelation is due to adaptation via sharpening, rather than alternative possibilities such as non-adaptation-related temporal hysteresis effects or attention. But this presents an interesting challenge for future research.

One very minor error:

p15, line 255: citation of "de Beeck et al..." (and corresponding entry in reference list at line 491) should be "Op de Beeck et al..."

Response to Reviewers

“Adaptation decorrelates shape representations”

Marcelo G. Mattar^{1*}, Maria Olkkonen^{2,3*}, Russell A. Epstein¹, & Geoffrey K. Aguirre⁴

1. Department of Psychology, University of Pennsylvania, Philadelphia, PA 19104, USA

2. Department of Psychology, Durham University, Durham, DH1 3LE, United Kingdom

3. Department of Psychology and Logopedics, Faculty of Medicine, University of Helsinki, Helsinki, 00100, Finland

4. Department of Neurology, University of Pennsylvania, Philadelphia, PA 19104, USA

* These authors contributed equally to this work

To the Reviewers,

We would like to thank you for evaluating our manuscript “Adaptation decorrelates shape representations” which we have revised in accordance with your suggestions and now resubmit for your consideration for publication in *Nature Communications*. In the pages that follow, we include our responses and the changes made point-by-point. For ease of reference, reviewer comments have been consecutively numbered and italicized.

1 Reviewer 1

1.1 *The decorrelation effect appears to be weak. This effect was tested on a relatively small sample of participants (n=10), and only a subset of these participants showed decorrelation. I appreciate that fMRI data from individual participants may not be significant; however Figure 3c demonstrates a rather weak effect of decorrelation as most data points are on the diagonal rather than shifted towards the y axis (only 3-4 data points appear to be shifted from the diagonal). Further the authors do not explain why the fMRI effect of decorrelation was significant only for stimulus space B but not space A. Replication of the effect across stimulus spaces would strengthen the evidence for this rather weak effect.*

We appreciate the reviewer’s desire to observe a significant effect for each individual participant and for each stimulus space. We did not find this. We find, however, the accumulated evidence in the study quite supportive of our central claim: (i) we show the effect behaviorally; (ii) we show an effect with fMRI that is in the predicted direction for all ten participants; (iii) this effect is statistically significant for one stimulus space but not the other, but the interaction is highly significant; (iv) our searchlight analysis reveals that the effect is present in object-selective areas of the visual cortex. We believe that, taken together, these results support the decorrelation hypothesis for shapes.

1.2 *The evidence provided by the authors regarding the link between fMRI and behavioral data is rather indirect. The authors write: ‘There is indeed a larger fMRI decorrelation effect for the stimulus space that exhibits a larger behavioral effect.’ However, the lack of both behavioral and fMRI data on the same individuals undermines the evidence for this link. I appreciate the constraints related to data collection; however, conclusions about neural mechanisms (i.e. decorrelation) that underlie behavior cannot be made without data-driven evidence.*

We agree that comparing the neural and behavioral effects in the same individuals would be ideal, and this is unfortunately a limitation of our study. The previous version of our manuscript was carefully edited to ensure that our contribution to these questions was clearly stated.

1.3 *The authors question the possible role of attention in decorrelation. However, several neurophysiology studies (e.g. Reynolds) provide evidence that attention decorrelates neural representations in the ventral visual stream (e.g. area V4). Again here, the lack of data undermines the conclusions of the study. The authors state that they did not collect participant responses for the attentional task. Therefore, it remains unclear whether the decorrelation observed relates to differences in attention or adaptation across stimulus conditions.*

Unfortunately, ruling out attention confounds is impossible in this case, as it would require us to ensure that attention is *not* modulated between conditions. The reviewer is right that the effect we report might be due to an attention modulation, but our stance is that attention effects are indistinguishable from what we term decorrelation.

1.4 *Finally, the manuscript fails to provide a mechanistic account and differentiate between mechanisms of decorrelation; that is, sharpening vs. fatigue. Instead the authors relate response suppression to decorrelation. This is interesting but it does not advance our understanding of the neural mechanisms that underlie adaptation.*

In a prior version of the manuscript, we explored if our data could adjudicate between a sharpening or fatigue mechanism of adaptation. As the reviewers noted at the time, our study is not able to gain sufficient traction on this question to be dispositive. Given that sharpening and fatigue are mechanisms described at the neural level, direct neural measurements might be needed.

2 Reviewer 2

2.1 *The authors have fully addressed all my concerns and I thank them for providing the additional data. I have no further comments. A very nice study.*

Thank you.

3 Reviewer 3

3.1 *Thank you to the authors for an extremely comprehensive and considered set of responses to the three reviewers. The revised manuscript is stronger, and the new repetition suppression analysis is interesting. As Reviewer 1 points out, since the decorrelation effect is small it would be great to replicate it in a larger set of observers and/or with different shape stimuli. I'm not sure this is a fatal flaw though, since the standard of evidence meets that of much other published literature.*

Thank you.

3.2 *I remain not fully convinced that the decorrelation is due to adaptation via sharpening, rather than alternative possibilities such as non-adaptation-related temporal hysteresis effects or attention. But this presents an interesting challenge for future research.*

We agree. As stated in Comment 1.4, the current study is unable to adjudicate between fatigue and sharpening. We have endeavored to make clear in our discussion section that our study does not resolve this interesting question.

3.3 *One very minor error: p15, line 255: citation of "de Beeck et al.." (and corresponding entry in reference list at line 491) should be "Op de Beeck et al.."*

Fixed.